# WarpGAN: Warping-Guided 3D GAN Inversion with Style-Based Novel View Inpainting

**Kaitao Huang[1], Yan Yan[1†], Jing-Hao Xue[2], Hanzi Wang[1]**

[1]Key Laboratory of Multimedia Trusted Perception and Efficient Computing,
Ministry of Education of China, Xiamen University, P.R. China
[2]Department of Statistical Science, University College London, UK
huangkt@stu.xmu.edu.cn, yanyan@xmu.edu.cn

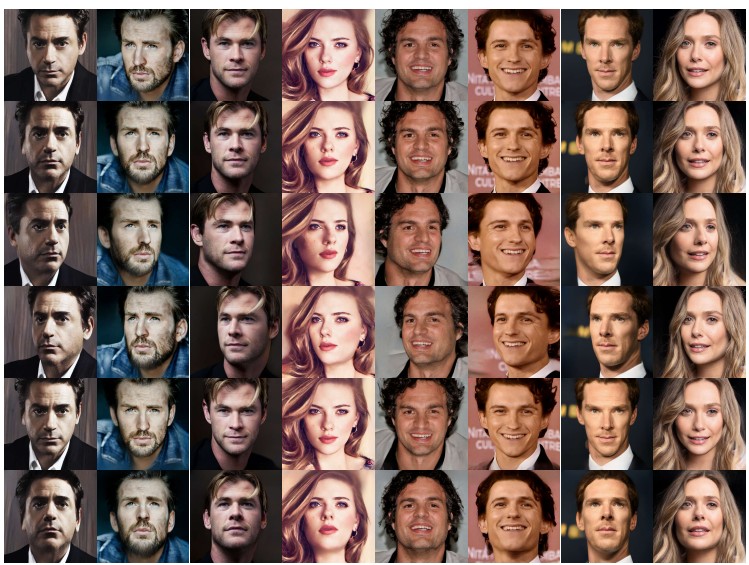

Figure 1: **Visual examples**. Given a single input image (the first row), our WarpGAN synthesizes images from five novel views: front, right, left, top, and down (the second to the sixth rows).

## Abstract

3D GAN inversion projects a single image into the latent space of a pre-trained 3D GAN to achieve single-shot novel view synthesis, which requires visible regions with high fidelity and occluded regions with realism and multi-view consistency. However, existing methods focus on the reconstruction of visible regions, while the generation of occluded regions relies only on the generative prior of 3D GAN. As a result, the generated occluded regions often exhibit poor quality due to the information loss caused by the low bit-rate latent code. To address this, we introduce the *warping-and-inpainting* strategy to incorporate image inpainting into 3D GAN inversion and propose a novel 3D GAN inversion method, **WarpGAN**. Specifically, we first employ a 3D GAN inversion encoder to project the single-view image into a latent code that serves as the input to 3D GAN. Then, we perform warping to a novel view using the depth map generated by 3D GAN. Finally, we develop a novel SVINet, which leverages the symmetry prior and multi-view image correspondence *w.r.t.* the same latent code to perform inpainting of occluded regions in the warped image. Quantitative and qualitative experiments demonstrate that our method consistently outperforms several state-of-the-art methods.

† Corresponding Author

Please visit the Project Page for visualizations and code.

39th Conference on Neural Information Processing Systems (NeurIPS 2025).

# 1 Introduction

GANs [13] have made remarkable progress in synthesizing unconditional images. In particular, StyleGAN [20, 21] has achieved photorealistic quality on high-resolution images. Several extensions [15, 31, 36] leverage the latent space (i.e., the $\mathcal{W}$ space) to control semantic attributes (e.g., expression and age). However, these 2D GANs suffer from inferior control over geometrical aspects of generated images, leading to multi-view inconsistency for viewpoint manipulation.

Recently, with the development of neural radiance fields (NeRF) [27] in novel view synthesis (NVS), a variety of 3D GANs [2, 5, 6, 14, 29, 39, 41] have been proposed to integrate NeRF into style-based generation, resulting in remarkable success in generating highly realistic images. Based on it, 3D GAN inversion methods project a single image into the latent space of a pre-trained 3D GAN generator, obtaining a latent code. Hence, the viewpoint of the input image can be changed by altering the camera pose, and the image attributes can be easily edited by modifying the latent code. Unlike 2D GAN inversion, 3D GAN inversion aims to generate images that maintain both the faithfulness of the input view and the high quality of the novel views.

On the one hand, existing 3D GAN inversion methods rely only on the generative prior of 3D GANs for generating the occluded regions (i.e., the invisible regions in the input image) in the novel viewpoint, resulting in unfaithful reconstruction of occluded regions in complex scenarios. On the other hand, for 3D scene generation, several recent methods adopt a *warping-and-inpainting* strategy. They [11, 30, 35] first predict a depth map of a given image, and then warp the input image to novel camera viewpoints with the depth-based correspondence, followed by a 2D inpainting network to synthesize high-fidelity occluded regions of the warped images.

To address the inferior reconstruction capability of occluded regions in existing 3D GAN inversion methods, motivated by the success of the *warping-and-inpainting* strategy in 3D scene generation, we introduce image inpainting into 3D GAN inversion. Unfortunately, 3D GAN inversion is dedicated to training with single-view datasets, while the above 3D scene generation methods usually require multi-view datasets for training. This leads to two issues: (1) **multi-view inconsistency** due to the lack of 3D information (i.e., the real novel view image) to guide the inpainting process; (2) **the unavailability of ground-truth images** from novel views to compute the loss during model training.

In this paper, we propose a novel 3D GAN inversion method, **WarpGAN**, by integrating the *warping-and-inpainting* strategy into 3D GAN inversion. Specifically, we first train a 3D GAN inversion encoder, which projects the input image into a latent code $w^+$ (located in the latent space $\mathcal{W}^+$ of the 3D GAN generator). By feeding $w^+$ into 3D GAN, we compute the depth map of the input image for geometric warping and perform an initial filling of the occluded regions in the warped image. Subsequently, leveraging the symmetry prior [43, 45] and multi-view image correspondence *w.r.t.* the same latent code in 3D GANs, we train a style-based novel view inpainting network (SVINet). It can inpaint the occluded regions in the warped image from the original view to the novel view. Hence, we can synthesize plausible novel view images with multi-view consistency. To address the unavailability of ground-truth images, we re-warp the image in the novel view back to the original view and feed it to SVINet. Hence, the loss can be calculated between the inpainting result and the input image. Some visual examples obtained by WarpGAN are given in Fig. 1.

In summary, the contributions of this paper are as follows:

- We propose a novel 3D GAN inversion method, WarpGAN, which successfully introduces the *warping-and-inpainting* strategy into 3D GAN inversion, substantially enhancing the quality of occluded regions in novel view synthesis.
- We introduce a style-based novel view inpainting network, SVINet, by fully leveraging the symmetry prior and the same latent code generated by 3D GAN inversion, achieving multi-view consistency inpainting on the occluded regions of warped images in novel views.
- We perform extensive experiments to validate the superiority of WarpGAN, showing the great potential of the *warping-and-inpainting* strategy in 3D GAN inversion.

# 2 Related work

**3D-Aware GANs.** Recent advancements in 3D-Aware GANs [2, 5, 6, 14, 29, 39, 41] effectively combine the high-quality 2D image synthesis of StyleGAN [20, 21] with the multi-view synthesis

capability of NeRF [27], advancing high-quality image synthesis from 2D to 3D and enabling multi-view image generation. These methods typically employ a two-stage generation pipeline, where a low-resolution raw image and feature maps are rendered, followed by upsampling to high-resolution using 2D CNN layers. Such a way ensures geometric consistency across multiple views and achieves impressive photorealism. In this paper, we leverage EG3D [5] as our 3D-aware GAN architecture, which introduces a hybrid explicit-implicit 3D representation (known as the tri-plane).

**GAN Inversion.** Although recent 2D GAN inversion methods [42] have achieved promising editing performance, they suffer from severe flickering and inevitable multi-view inconsistency when editing 3D attributes (e.g., head pose) since the pretrained generator is not 3D-aware. Hence, 3D GAN inversion is developed to maintain multi-view consistency when rendering novel viewpoints. However, directly transferring 2D methods to 3D without effectively incorporating 3D information will inevitably lead to geometry collapse and artifacts.

Similar to 2D GAN inversion, 3D GAN inversion can be categorized into optimization-based methods and encoder-based methods. Some optimization-based methods [23, 43, 45] generate multiple pseudo-images from different viewpoints to facilitate optimization. For instance, HFGI3D [43] leverages visibility analysis to achieve pseudo-multi-view optimization; SPI [45] utilizes the facial symmetry prior to synthesize pseudo multi-view images; and Pose Opt. [23] simultaneously optimizes camera pose and latent codes. In addition, In-N-Out [44] optimizes a triplane for out-of-distribution object reconstruction and employs composite volume rendering. Encoder-based methods project the input image into the latent space of the 3D GAN generator and then employ the generative capacity of the 3D GAN to synthesize novel-view images, while fully utilizing the input image to reconstruct the visible regions of the novel-view images. For example, GOAE [46] computes the residual between the input image and the reconstructed image to complement the $\mathcal{F}$ space of the generator, and introduces an occlusion-aware mix tri-plane for novel-view image generation; Triplanenet [3] calculates an offset for the triplane based on the residual and proposes a facial symmetry prior loss; and Dual Encoder [4] employs two encoders (one for visible regions and the other for occluded regions) for inversion and introduces an occlusion-aware triplane discriminator to enhance both fidelity and realism.

Our method is intrinsically different from existing methods that rely heavily on 3D GAN generative priors to generate occluded regions. Our method introduces a novel inpainting network to fill the occluded regions, facilitating the generation of rich details.

**Depth-based Warping for Single-shot Novel View Synthesis.** Some 3D GAN inversion methods [23, 43, 45] use depth-based warping to synthesize pseudo multi-view images for optimization. SPI [45] warps the input image to an adjacent view for pseudo-supervision. Pose Opt. [23] warps the image from the canonical viewpoint to the input viewpoint to assist training. HFGI3D [43] utilizes a 3D GAN to fill the occluded regions of the warped image from the input view to novel views, synthesizing several pseudo novel-view images. However, these methods only rely on a 3D GAN to generate occluded regions, failing to achieve satisfactory results in occluded regions under complex scenarios.

Recently, some methods follow the *warping-and-inpainting* strategy on single-shot NVS for general scenes [11, 30, 35]. They first predict a depth map for the input image, then warp the input image to a novel view using the depth map, and finally perform inpainting on the occluded regions in the novel view. This way can effectively preserve the information of the input image while leveraging the powerful inpainting capability of 2D inpainting networks to generate reasonable content for occluded regions. Inspired by this strategy, we introduce a 2D inpainting network into 3D GAN inversion by effectively exploiting the symmetry prior and the latent code of the input image.

## 3 Methodology

### 3.1 Overview

As shown in Fig. 2, our WarpGAN consists of a 3D GAN inversion network (including a 3D GAN inversion encoder and a 3D-aware GAN) and a style-based novel view inpainting network (SVINet). First, we utilize a 3D GAN inversion encoder $E_{w+}$ to project the input image $\mathbf{I}$ into the latent space $\mathcal{W}^+$ of the 3D GAN generator, obtaining the latent code $w^+$. Based on this, we utilize a rendering decoder to render the depth map $\mathbf{D}$ of $\mathbf{I}$ and the novel view image $\hat{\mathbf{I}}_{novel}^{w^+}$. Under the guidance of the depth map $\mathbf{D}$, we warp the input image $\mathbf{I}$ from the original view $c$ to the novel view $c_{novel}$, thereby obtaining the warped image $\mathbf{I}_{c \to c_{novel}}^{warp}$ and the occluded regions $\mathbf{M}_{c \to c_{novel}}^{o}$ of the input image in the

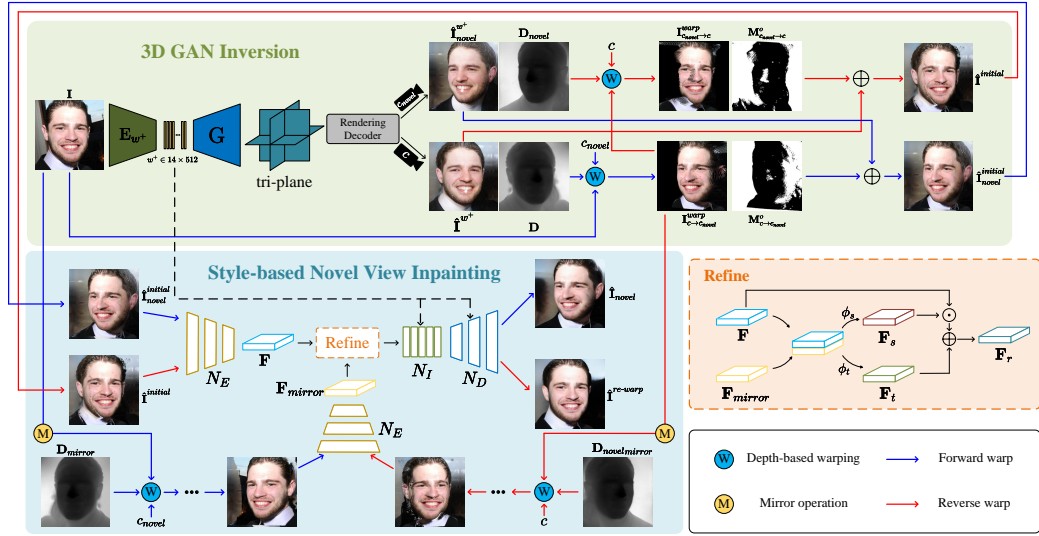

Figure 2: **Overview of our WarpGAN**, which consists of a 3D GAN inversion network and a style-based novel view inpainting network (SVINet). The "Forward warp" flow (blue arrows) illustrates the inference process of novel view synthesis. During model training, we also require the "Reverse warp" flow (red arrows) to warp the novel view image back to the original view for loss computation.

target view, that is,

$$\mathbf{I}^{warp}_{c \to c_{novel}}, \mathbf{M}^{o}_{c \to c_{novel}} = \text{warp}(\mathbf{I}; \mathbf{D}, \pi_{c \to c_{novel}}, K), \tag{1}$$

where $\pi_{c \to c_{novel}}$ is a relative camera pose between $c$ and $c_{novel}$, $K$ is the camera intrinsic matrix, and warp$(\cdot)$ is a geometric warping function [28, 35] which unprojects pixels of the input image $\mathbf{I}$ with its depth map $\mathbf{D}$ to the 3D space, and reprojects them based on $\pi_{c \to c_{novel}}$ and $K$.

Then, we use $\hat{\mathbf{I}}^{w^+}_{novel}$ to fill in the occluded regions of $\mathbf{I}^{warp}_{c \to c_{novel}}$, serving as the initial result $\hat{\mathbf{I}}^{initial}_{novel}$ for the occluded regions, which can be formulated as

$$\hat{\mathbf{I}}^{initial}_{novel} = \mathbf{I}^{warp}_{c \to c_{novel}} + \mathbf{M}^{o}_{c \to c_{novel}} \cdot \hat{\mathbf{I}}^{w^+}_{novel}. \tag{2}$$

Subsequently, the initial result $\hat{\mathbf{I}}^{initial}_{novel}$ is fed into SVINet for further inpainting, giving the final output $\hat{\mathbf{I}}_{novel}$ of WarpGAN. Notably, we employ symmetry-aware feature extraction and modulate the convolutions of the inpainting network with $w^+$ during the inpainting process. We also construct a style-based loss to ensure consistency between the generated image in the novel view and the original view image.

### 3.2 3D GAN Inversion Encoder

Similar to existing encoder-based 3D GAN inversion methods, our 3D GAN inversion encoder $E_{w^+}$ projects an input image $\mathbf{I}$ with the camera pose $c$ into the latent space $\mathcal{W}^+$ of the pre-trained 3D GAN, obtaining the latent code $w^+ = E_{w^+}(\mathbf{I})$. Then, we leverage the generator $G(\cdot)$ of 3D GAN to generate the tri-plane and use the rendering decoder $\mathcal{R}$ to render images at specified camera poses. Based on above, we perform image reconstruction $\hat{\mathbf{I}}^{w^+} = \mathcal{R}(G(w^+), c)$ by specifying the camera pose as $c$. In this way, we obtain the novel view image $\hat{\mathbf{I}}^{w^+}_{novel}$ corresponding to the novel camera pose $c_{novel}$. Under the principles of NeRF, we replace the color of the sampling points with the distance to the camera during the rendering process, obtaining the depth maps $\mathbf{D}$ and $\mathbf{D}_{novel}$. More implementation details can be found in the **Appendix**.

Inspired by GOAE [46], we employ a pyramid-structured Swin-Transformer [26] as the backbone of the encoder, based on which we leverage feature layers at different scales to generate latent codes at various levels.

Since our dataset contains only single-view images, we train $E_{w+}$ using a reconstruction loss $\mathcal{L}_{w+}$, which includes a pixel-wise (MSE) loss $\mathcal{L}_2$, a perceptual loss $\mathcal{L}_{\text{LPIPS}}$ [48], and an identity loss $\mathcal{L}_{\text{ID}}$ with a pre-trained ArcFace network [12]:

$$\mathcal{L}_{w+}(\hat{\mathbf{I}}^{w^+}, \mathbf{I}) = \lambda_2 \mathcal{L}_2(\hat{\mathbf{I}}^{w^+}, \mathbf{I}) + \lambda_{\text{LPIPS}} \mathcal{L}_{\text{LPIPS}}(\hat{\mathbf{I}}^{w^+}, \mathbf{I}) + \lambda_{\text{ID}}^{w^+} \mathcal{L}_{\text{ID}}(\hat{\mathbf{I}}^{w^+}, \mathbf{I}), \tag{3}$$

where $\lambda_2$, $\lambda_{\text{LPIPS}}$, and $\lambda_{\text{ID}}^{w^+}$ denote the loss weights for $\mathcal{L}_2$, $\mathcal{L}_{\text{LPIPS}}$, and $\mathcal{L}_{\text{ID}}$, respectively.

### 3.3 Style-Based Novel View Inpainting Network (SVINet)

Due to the existence of occluded regions in the novel view, the warped image contains "holes" (see Fig. 2 for an illustration). To generate high-quality novel-view images, we propose a style-based novel view inpainting network (SVINet) to fill in the "holes" in the warped image.

As shown in Fig. 2, our SVINet follows the traditional "encode-inpaint-decoder" architecture [10, 24, 37], consisting of three sub-networks: $N_E$, $N_I$, and $N_D$. Technically, $N_E$ is first used to extract features from the model input while performing downsampling. Then, the inpainting operation is performed in the feature space by using $N_I$. Finally, $N_D$ is used to upsample the features to obtain the inpainted image.

#### 3.3.1 Symmetry-Aware Feature Extraction

We first use the novel-view image $\hat{\mathbf{I}}_{novel}^{w^+}$ obtained from 3D GAN inversion to fill in the occluded regions in the warped image $\mathbf{I}_{c \rightarrow c_{novel}}^{warp}$ (Eq. (1)), resulting in an initial inpainting result $\hat{\mathbf{I}}_{novel}^{initial}$ (Eq. (2)). We then feed $\hat{\mathbf{I}}_{novel}^{initial}$ into $N_E$ to obtain the feature $\mathbf{F}$. In addition, we also propose to leverage the facial symmetry [43, 45] by warping the mirrored input image $\mathbf{I}_{mirror}$ to the target view $c_{novel}$, obtaining $\mathbf{I}_{mirror}{}_{c_{mirror} \rightarrow c_{novel}}^{warp}$. The mirrored image is then processed in the same manner as described above and fed into $N_E$ to obtain the mirror feature $\mathbf{F}_{mirror}$.

Subsequently, we utilize $\mathbf{F}$ and $\mathbf{F}_{mirror}$ to predict the scale map $\mathbf{F}_s$ and the translation map $\mathbf{F}_t$, which can be used to refine $\mathbf{F}$ via featurewise linear modulation (FiLM) [32], obtaining $\mathbf{F}_r$, that is,

$$\begin{aligned} \{\mathbf{F}_s, \mathbf{F}_t\} &= \{\phi_s([\mathbf{F}, \mathbf{F}_{mirror}]_1), \phi_t([\mathbf{F}, \mathbf{F}_{mirror}]_1)\}, \\ \mathbf{F}_r &= \mathbf{F}_s \odot \mathbf{F} + \mathbf{F}_t, \end{aligned} \tag{4}$$

where $\phi_s$ and $\phi_t$ are convolutional neural networks; $[,]_1$ denotes concatenation along the 1th dimension, i.e., the channel dimension; '$\odot$' denotes the Hadamard product.

Next, $\mathbf{F}^r$ is successively fed into $N_I$ and $N_D$ to obtain the inpainting result $\hat{\mathbf{I}}_{novel}$.

#### 3.3.2 Style-Based Inpainting

Inpainting networks typically rely on the information of the input image to fill in the missing regions. However, due to the limited information contained in single-view images, using only this information for inpainting may lead to the issue of multi-view inconsistency. To address the consistency issue, motivated by the fact that images of the same object from different viewpoints share the same latent code in 3D GANs, we introduce the latent code to control the image inpainting process.

Technically, we modulate the convolutions [21, 24] in the "inpaint" and "decoder" parts of the inpainting network using the latent code $w^+$ obtained from $E_{w+}$. This modulation of the convolutions facilitates us to control the inpainting process for occluded regions, achieving multi-view consistency in the generated images.

Specifically, we first employ a mapping function $\mathcal{A}$ to obtain the style code $s = \mathcal{A}(w^+)$. Then the weights of the convolutions $w$ are modulated as

$$\begin{aligned} w'_{ijk} &= s_i \cdot w_{ijk}, \\ w''_{ijk} &= w'_{ijk} \Big/ \sqrt{\sum\nolimits_{i,k} {w'_{ijk}}^2 + \epsilon}, \end{aligned} \tag{5}$$

where $w''$ denotes the final modulated weights; $s_i$ is the scale corresponding to the $i$th input feature map; $j$ and $k$ enumerate the output feature maps and spatial footprint of the convolution, respectively.

### 3.3.3 Training strategy

**Real data.** Since our real dataset contains only single-view images, no target-view images can be used to compute the loss and update the model parameters when synthesizing images from novel views. To address this, we propose to re-warp the warped image from the novel view back to the original view, and then compute the loss between the inpainting result and the input image.

Specifically, for the input image $\mathbf{I}$, we first warp it to the novel view $c_{novel}$ to obtain $\mathbf{I}^{warp}_{c \to c_{novel}}$, and then inpaint it using SVINet to get $\hat{\mathbf{I}}_{novel}$. Next, we re-warp $\mathbf{I}^{warp}_{c \to c_{novel}}$ back to the source view $c$ and inpaint it again to obtain $\hat{\mathbf{I}}^{re-warp}$. Based on the above, given the input image $\mathbf{I}$, we obtain two inpainted images $\hat{\mathbf{I}}_{novel}$ and $\hat{\mathbf{I}}^{re-warp}$ for loss computation.

**Synthetic data.** In addition to real data, we also utilize synthetic data to assist in training our model. We sample a latent code $w_{synth}$ from the latent space of 3D GAN and generate two images $\mathbf{I}^{synth}_s$ and $\mathbf{I}^{synth}_t$ from different viewpoints. We then warp $\mathbf{I}^{synth}_s$ from the source view to the target view and input it into SVINet to obtain the inpainted image $\hat{\mathbf{I}}^{synth}_t$. Finally, we compute the loss between $\hat{\mathbf{I}}^{synth}_t$ and $\mathbf{I}^{synth}_t$.

**Loss function.** Our loss function consists of three components: the reconstruction loss, the consistency loss, and the adversarial loss. The reconstruction loss $\mathcal{L}_{rec}$ includes the pixel-wise MAE loss $\mathcal{L}_1$, the perceptual loss $\mathcal{L}_P$ [37], and the identity loss $\mathcal{L}_{ID}$ [12]:

$$\mathcal{L}_{rec}(\hat{\mathbf{I}}, \mathbf{I}) = \lambda_1 \mathcal{L}_1(\hat{\mathbf{I}} - \mathbf{I}) + \lambda_P \mathcal{L}_P(\hat{\mathbf{I}}, \mathbf{I}) + \lambda_{ID} \mathcal{L}_{ID}(\hat{\mathbf{I}}, \mathbf{I}), \tag{6}$$

where $\lambda_1$, $\lambda_P$, and $\lambda_{ID}$ denote the loss weights for $\mathcal{L}_1$, $\mathcal{L}_P$, and $\mathcal{L}_{ID}$, respectively; $\hat{\mathbf{I}}$ and $\mathbf{I}$ represent the input image and the generated image, respectively.

To ensure multi-view consistency, we introduce the consistency loss $\mathcal{L}_c$, which computes the MSE between the latent codes of the original image and the inpainted image. This loss is used to control the multi-view consistency of the generated images:

$$\mathcal{L}_c(\hat{\mathbf{I}}, \mathbf{I}) = ||\mathbf{E}_{w+}(\hat{\mathbf{I}}), \mathbf{E}_{w+}(\mathbf{I})||_2. \tag{7}$$

To further enhance the quality of the inpainted images, we also use an adversarial loss:

$$\mathcal{L}^G_{adv} = -\mathbb{E}[\log(D(\hat{x}))], \tag{8}$$

$$\mathcal{L}^D_{adv} = -\mathbb{E}[\log(D(x))] - \mathbb{E}[\log(1 - D(\hat{x}))] + \gamma \mathbb{E}[||\nabla D(x)||_2], \tag{9}$$

where $x$ denotes the real and synthetic images (i.e., $\mathbf{I}$ and $\mathbf{I}^{synth}_t$); $\hat{x}$ represents the inpainted images (i.e., $\hat{\mathbf{I}}_{novel}$, $\hat{\mathbf{I}}^{re-warp}$, and $\hat{\mathbf{I}}^{synth}_t$); $D$ denotes the discriminator [10, 24, 37].

In summary, the loss function for SVINet can be formulated as follows:

$$\begin{aligned} \mathcal{L}_{SVINet} &= \lambda_{rec} \mathcal{L}_{rec}([\hat{\mathbf{I}}^{re-warp}, \hat{\mathbf{I}}^{synth}_t]_0, [\mathbf{I}, \mathbf{I}^{synth}_t]_0) \\ &+ \lambda_c \mathcal{L}_c([\hat{\mathbf{I}}_{novel}, \hat{\mathbf{I}}^{re-warp}, \hat{\mathbf{I}}^{synth}_t]_0, [\mathbf{I}, \mathbf{I}, \mathbf{I}^{synth}_t]_0) + \lambda_{adv} \mathcal{L}^G_{adv}, \end{aligned} \tag{10}$$

where $[,]_0$ denotes concatenation along the 0-th dimension (i.e., the batch dimension); $\lambda_{rec}$, $\lambda_c$, and $\lambda_{adv}$ denote the loss weights for $\mathcal{L}_{rec}$, $\mathcal{L}_c$, and $\mathcal{L}^G_{adv}$, respectively.

## 4 Experiments

### 4.1 Experimental Settings

**Datasets.** Our experiments mainly focus on face datasets. We use the FFHQ dataset [20] and 100K pairs of synthetic data for training. The synthetic pairs $\{\mathbf{I}^{synth}_s, \mathbf{I}^{synth}_t\}$ are generated from EG3D [5], sharing the same latent code $w_{synth}$ but rendered with different camera poses. To evaluate the generalization ability of our method, we employ the CelebA-HQ dataset [19] and the multi-view MEAD dataset [40] for testing. We preprocess the images in the datasets and extract their camera poses in the same manner as [5].

**Implementation Details.** For all experiments, we employ the EG3D [5] generator pre-trained on FFHQ. For the 3D GAN inversion encoder $\mathbf{E}_{w+}$, we set the batch size to 4 and train it for 500K

Table 1: **Comparisons with state-of-the-art methods** on the CelebA-HQ and MEAD datasets.

| Category | Method | CelebA-HQ | | MEAD | | | | | | Time (s)↓ |
|---|---|---|---|---|---|---|---|---|---|---|
| | | FID↓ | ID↑ | LPIPS ↓ | | FID ↓ | | ID ↑ | | |
| | | | | ±30° | ±60° | ±30° | ±60° | ±30° | ±60° | |
| Optimization | SG2 $\mathcal{W}^+$ | 26.09 | 0.7369 | 0.2910 | 0.3372 | 39.30 | 64.47 | 0.7992 | 0.7533 | 43.72 |
| | PTI | 25.70 | 0.7616 | 0.2771 | 0.3341 | 44.23 | 66.00 | 0.8089 | 0.7582 | 62.65 |
| | Pose Opt. | 29.04 | 0.7500 | 0.2990 | 0.3428 | 52.25 | 73.23 | 0.7954 | 0.7405 | 91.60 |
| | HFGI3D | 24.30 | 0.7641 | 0.2775 | 0.3494 | 51.24 | 79.81 | 0.8019 | 0.7370 | 264.5 |
| Encoder | pSp | 38.46 | 0.7375 | 0.3116 | 0.3720 | 65.21 | 94.34 | 0.7900 | 0.7401 | 0.05430 |
| | GOAE | 35.41 | 0.7498 | 0.2818 | 0.3453 | 59.69 | 86.23 | 0.8109 | 0.7370 | 0.07999 |
| | Triplanenet | 32.65 | 0.7706 | 0.3379 | 0.4103 | 76.62 | 130.55 | 0.8059 | 0.7135 | 0.1214 |
| | Ours | **19.12** | **0.7882** | **0.2490** | **0.3008** | **38.15** | **64.01** | **0.8315** | **0.7741** | 0.08390 |

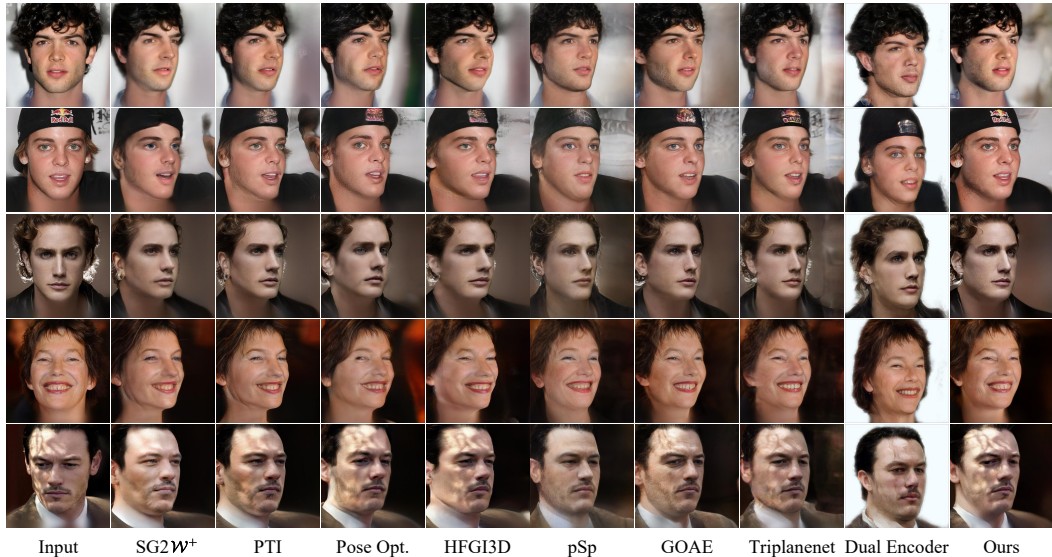

Input SG2$\mathcal{W}^+$ PTI Pose Opt. HFGI3D pSp GOAE Triplanenet Dual Encoder Ours

Figure 3: **Comparisons of novel view synthesis** on the CelebA-HQ dataset between our WarpGAN and several state-of-the-art methods.

iterations on the FFHQ dataset. We use the Ranger optimizer, which combines Rectified Adam [25] with the Lookahead technique [47], with learning rates of 1e-4 for $E_{w^+}$. The values of $\lambda_2$, $\lambda_{\text{LPIPS}}$, and $\lambda_{\text{ID}}^{w^+}$ in Eq. (3) are set to 1.0, 0.8, and 0.1. For SVINet, we set the batch size to 2 and train it for 300K iterations on both the FFHQ dataset and synthetic data pairs. For the novel view camera poses during the training process, we sample from the camera poses of the pose-rebalanced FFHQ dataset [5]. We use the Adam optimizer [22], with learning rates of 1e-3 and 1e-4 for the SVINet and discriminator, respectively. The values of $\lambda_1$, $\lambda_P$, and $\lambda_{\text{ID}}$ in Eq. (6) are set to 10.0, 30.0, and 0.1, respectively. The values of $\lambda_{\text{rec}}$, $\lambda_c$, and $\lambda_{\text{adv}}$ in Eq. (10) are set to 1.0, 0.1, and 10.0, respectively.

**Baselines.** We compare our WarpGAN with several 3D GAN inversion methods, including optimization-based methods (such as SG2 $\mathcal{W}^+$ [1], PTI [34], Pose Opt. [23], and HFGI3D [43]) and encoder-based methods (such as pSp [33], GOAE [46], Triplanenet [3], and Dual Encoder [4]). Note that Dual Encoder employs a 3D GAN other than EG3D and removes the background during training. This is different from our experimental setup, we only compare it in the qualitative analysis.

**Evaluation metrics.** We perform novel view synthesis evaluation on the CelebA-HQ dataset and the MEAD dataset. For the CelebA-HQ dataset, we compute the Fréchet Inception Distance (FID) [17] and ID similarity [12] between the original images and the novel view images. For the multi-view MEAD dataset, each person includes five face images with increasing yaw angles (front, ±30°, and ±60°). We use the front image as input and synthesize the other four views. We then compute the LPIPS [48], FID, and ID similarity between the synthesized images and their corresponding ground-truth images. The inference times (Time) in Table 1 are measured on a single Nvidia GeForce RTX 4090 GPU.

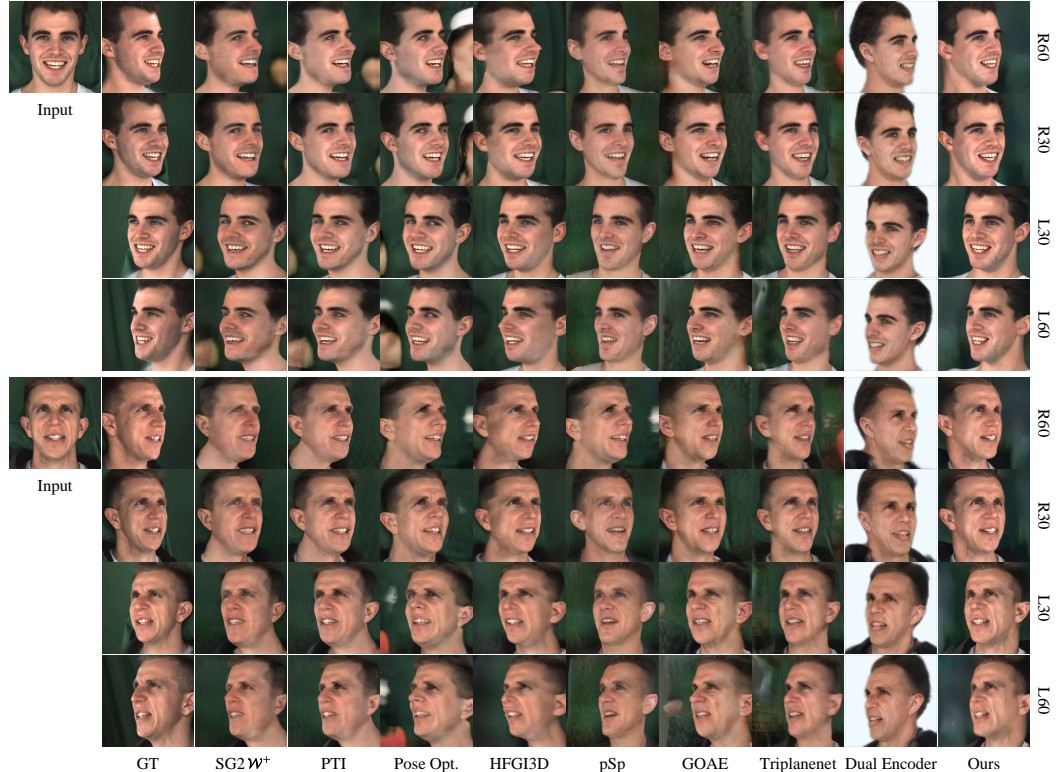

Figure 4: **Comparisons of different methods** on the MEAD dataset for synthesizing images of the other four views (R60, R30, L30, and L60) using the front image as input.

## 4.2 Comparisons with State-of-the-Art Methods

**Quantitative Evaluation.** As shown in Table 1, we provide the performance of different methods on the CelebA-HQ dataset and the MEAD dataset. It can be clearly observed that optimization-based methods achieve better performance than encoder-based methods, but at the cost of significantly higher inference times. Among them, HFGI3D, which performs optimization twice using PTI (once for filling the occluded regions of warped images and once for multi-view optimization), shows substantial performance improvement but suffers from slow inference times. In contrast, our WarpGAN, which has an inference time comparable to encoder-based methods, surpasses the performance of optimization-based methods. The excellent performance on the MEAD dataset demonstrates that our method is capable of effectively preserving multi-view consistency.

**Qualitative Evaluation.** We provide visualization results of novel view synthesis in Fig. 3 and Fig. 4. By successfully integrating the *warping-and-inpainting* strategy into 3D GAN inversion, our method can better preserve facial details and generate more reasonable occluded regions. Moreover, our method is capable of maintaining 3D consistency in novel views more naturally.

## 4.3 Ablation Studies

Table 2: **Ablation** on different components of our WarpGAN.

| Name | Model | FID $\downarrow$ | ID $\uparrow$ |
|------|-------|------|------|
| A | $E_{w+}$ | 36.07 | 0.7437 |
| B | w/o SVINet | 29.28 | 0.7735 |
| C | w/o Mod$_{w+}$ & $\mathcal{L}_c$ | 19.71 | 0.7879 |
| D | w/o Mod$_{w+}$ | 19.47 | 0.7880 |
| E | w/o symmetry | 20.04 | 0.7825 |
| F | w/o synth data | 19.18 | 0.7880 |
| G | Full Model | **19.12** | **0.7882** |

To investigate the contributions of key components in our method, we conduct ablation studies. In Table 2, we compare the quality of novel view synthesis using different model variants on the CelebA-HQ dataset.

Comparing "B" and "G" clearly demonstrates the significant role of SVINet in inpainting occluded regions. Comparing "C", "D", and "G" shows that modulating the convolutions of SVINet with $w^+$ and incorporating $\mathcal{L}_c$ enhance the performance of our method. Comparing "E"

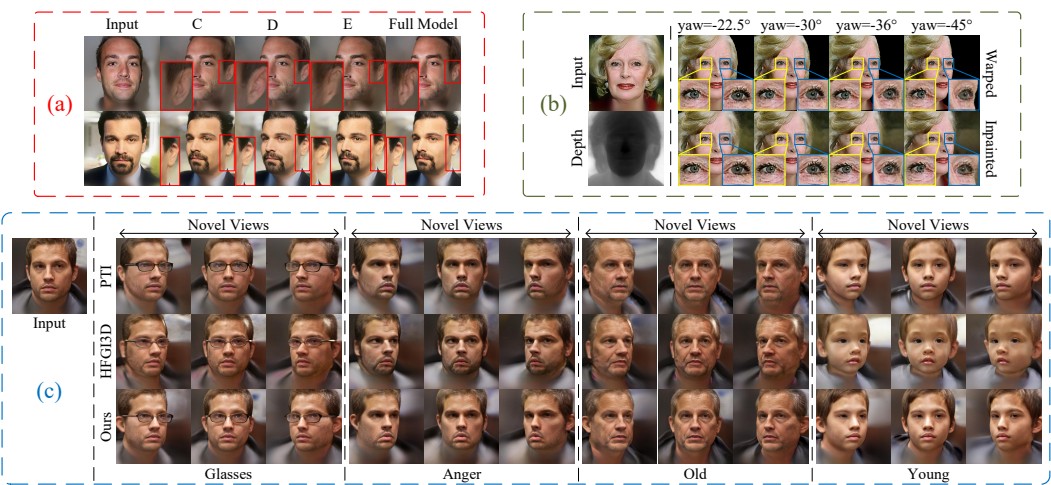

Figure 5: (a) **Qualitative comparisons** of the Full Model with model variants "C", "D", and "E"; (b) **Some failure cases**; (c) **Comparisons of image attribute editing effects** with PTI and HFGI3D.

and "G" indicates that leveraging facial symmetry prior helps generate occluded regions in novel views. Comparing "F" and "G" reveals that training with synthetic data slightly improves the quality of novel view synthesis. We also qualitatively compare "C", "D", "E", and "G" (Full Model) in Fig. 5(a). Incorporating the latent code to control the inpainting process of SVINet and the symmetry prior can provide more information, reduce blurring and artifacts, and generate more detailed results.

### 4.4 Editing Application

Since our WarpGAN achieves novel view synthesis by inpainting warped images, the visible parts of the novel view images are minimally affected by the latent code. Consequently, manipulating the latent code alone does not enable attribute editing of the image. To address this issue, similar to HFGI3D [43], we utilize WarpGAN to synthesize a series of novel view images, which are then fed into PTI [34] for optimization. This process yields an optimized latent code $w_{opt}^+$ and a fine-tuned 3D GAN generator. In this way, attribute editing of the input image and novel view rendering can be achieved by editing $w_{\text{opt}}^+$ [15, 31, 36] and modifying the camera pose $c$. As shown in Fig. 5(c), we perform attribute editing on the input image for four attributes: "Glasses", "Anger", "Old", and "Young", and compare the results with those from PTI and HFGI3D. It can be observed that the edited images obtained by using multi-view images synthesized by WarpGAN for optimization assistance exhibit higher fidelity and appear more natural.

## 5 Conclusion

In this paper, motivated by the achievement of the *warping-and-inpainting* strategy in 3D scene generation, we successfully integrate image inpainting with 3D GAN inversion and propose a novel 3D GAN inversion method, WarpGAN, for high-quality novel view synthesis from a single image. Our WarpGAN consists of a 3D GAN inversion network and SVINet. Specifically, we first obtain the depth of the input image using 3D GAN inversion, then apply depth-based warping to the input image to obtain the warped image, and finally use SVINet to fill in the occluded regions of the warped image. Notably, our SVINet leverages symmetry prior and the latent code for multi-view consistency inpainting. Extensive qualitative and quantitative experiments demonstrate that our method outperforms existing state-of-the-art optimization-based and encoder-based methods.

**Limitations.** Due to the inevitable errors in the depth map [11, 30, 35], the warped image sometimes become unreliable, which in turn prevents our SVINet from eliminating such artifacts. As illustrated in Fig. 5(b), when the angle variation is small, SVINet can alleviate the deformation of the eyes. However, as the angle of change increases, the output of SVINet deteriorates.

## Acknowledgments and Disclosure of Funding

This work was supported by the National Natural Science Foundation of China under Grant 62372388 and Grant U21A20514, the Major Science and Technology Plan Project on the Future Industry Fields of Xiamen City under Grant 3502Z20241029 and Grant 3502Z20241027, and the Fundamental Research Funds for the Central Universities under Grant 20720240076 and Grant ZYGX2021J004.

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

# A  Additional Architecture Details

**Detailed Structure of SVINet.** LaMa [37] introduces fast Fourier convolutions (FFCs) [8] into image inpainting, achieving a receptive field that covers the whole image even in the early network layers. Such a way can facilitate the inpainting of large missing areas. To effectively fill in the occluded regions of the warped image, our SVINet is built upon the framework of LaMa and consists of three sub-networks: $N_E$, $N_I$, and $N_D$. For an input image with the size of $512 \times 512$, $N_E$ includes 3 downsampling convolutional layers that downsample the input image to a feature map with the size of $64 \times 64$; $N_I$ contains 9 FFC residual blocks, each of which consists of two FFCs and a residual connection, for inpainting; and $N_D$ consists of 3 upsampling convolutional layers to upsample the image resolution back to the size of $512 \times 512$. The convolutions in $N_I$ and $N_D$ are modulated by the latent code $w^+$ from the 3D GAN inversion encoder $E_{w^+}$. Note that, each FFC contains three convolutional branches and one spectral transform branch, and the convolutions within the spectral transform are also modulated, as shown in Fig. 6.

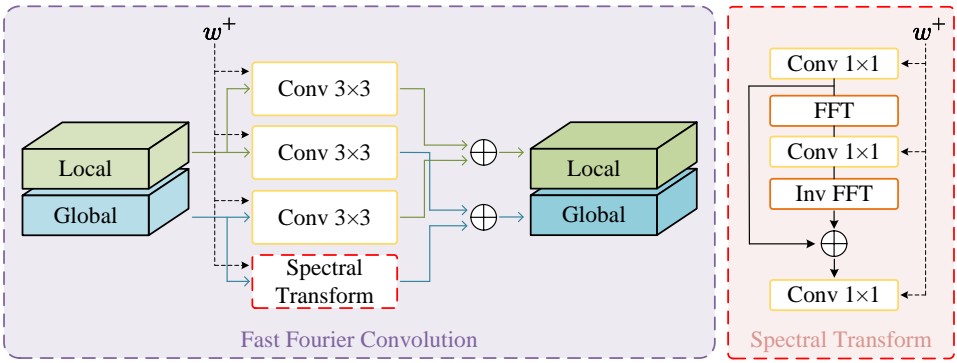

Figure 6: **The detailed structure** of fast Fourier convolution modulated by the latent code $w^+$.

# B  Additional Implementation Details

## B.1  Principles of Neural Radiance Fields

Neural Radiance Fields (NeRF) [27] employs a fully-connected deep network, which maps a 3D spatial location $\mathbf{x}$ and a viewing direction $\mathbf{d}$ to color $\mathbf{c}$ and density $\sigma$, to represent a scene. By querying $\mathbf{x}$ and $\mathbf{d}$ along camera rays and applying classical volume rendering techniques [18], the color and density information can be projected into a 2D image. Specifically, for each projected ray $\mathbf{r}$ corresponding to a given pixel, $N_s$ points (denoted as $\{t_i\}_{i=1}^{N_s}$) are sampled along the ray. For each sampled point, the estimated color and density are represented as $\mathbf{c}_i$ and $\sigma_i$, respectively. The RGB value $C(\mathbf{r})$ for each ray can then be computed via volumetric rendering as follows:

$$C(\mathbf{r}) = \sum_{i=1}^{N_s} T_i(1 - \exp(-\sigma_i \delta_i))\mathbf{c}_i, \tag{11}$$

where $T_i = \exp(-\sum_{j=1}^{i-1} \sigma_j \delta_j)$, and $\delta_i = t_{i+1} - t_i$ denotes the distance between adjacent samples.

Similarly, if we replace the color $c_i$ of each sampled point with the distance $t_i$ from the sampling point to the camera during volumetric rendering, the depth $d(\mathbf{r})$ along each ray can be obtained as

$$d(\mathbf{r}) = \sum_{i=1}^{N_s} T_i(1 - \exp(-\sigma_i \delta_i))t_i. \tag{12}$$

## B.2 Multi-View Optimization for Editing

Our WarpGAN synthesizes novel view images not only based on the results of 3D GAN inversion but also relies on the warping results of the input image. Thus, only modifying the latent code within our method is difficult to achieve desirable editing effects. Inspired by HFGI3D [43], we employs WarpGAN to generate $N$ novel view images $\{\mathbf{I}_i\}_{i=1}^N$ corresponding to $N$ different camera poses $\{c_i\}_{i=1}^N$ to assist the optimization process of PTI [34], denoted as **WarpGAN-Opt**.

Specifically, for a single input image $\mathbf{I}$ with the camera pose $c$, we first employ an optimization-based GAN inversion method [1] to jointly optimize the latent code $w^+$ and the noise vector $n$ in the 3D GAN generator:

$$w_{opt}^+, n = \underset{w^+, n}{\arg\min} \; \mathcal{L}_2(\mathcal{R}(\mathrm{G}(w^+, n; \theta), c), \mathbf{I}) + \lambda_n \mathcal{L}_n(n), \tag{13}$$

where $\mathcal{L}_n$ is a noise regularization term and $\lambda_n$ is a hyperparameter [34].

Subsequently, we fix the optimized latent code $w_{opt}^+$ and fine-tune the 3D GAN generator based on the input image $\mathbf{I}$ and a series of novel view images synthesized by our WarpGAN:

$$\theta^* = \underset{\theta}{\arg\min} \; \mathcal{L}_{\mathrm{G}}(\mathcal{R}(\mathrm{G}(w_{opt}^+; \theta), c), \mathbf{I}) + \lambda_{mv} \sum_i^N \mathcal{L}_{\mathrm{G}}(\mathcal{R}(\mathrm{G}(w_{opt}^+; \theta), c_i), \mathbf{I}_i), \tag{14}$$

$$\mathcal{L}_{\mathrm{G}} = \lambda_2^{\mathrm{G}} \mathcal{L}_2 + \lambda_{\mathrm{LPIPS}}^{\mathrm{G}} \mathcal{L}_{\mathrm{LPIPS}}, \tag{15}$$

where $\lambda_{mv}$ is set to 1.0; both $\lambda_2^{\mathrm{G}}$ and $\lambda_{\mathrm{LPIPS}}^{\mathrm{G}}$ are set to 1.0.

After the aforementioned process, we obtain the optimized latent code $w_{opt}^+$ and the 3D GAN generator with tuned weights $\theta^*$. To generate attribute-edited images from different viewpoints, we simply modify $w_{opt}^+$ [31, 36], specify the desired camera pose $c_{novel}$, and feed them into the 3D GAN to obtain the edited image $\hat{\mathbf{I}}_{novel}^{edit}$ in the novel view, that is,

$$\hat{\mathbf{I}}_{novel}^{edit} = \mathcal{R}(\mathrm{G}(w_{opt}^+ + \alpha \mathbf{n}_{att}; \theta^*), c_{novel}), \tag{16}$$

where $\mathbf{n}_{att}$ denotes a specific direction for attribute editing and $\alpha$ is a scaling factor.

## C  Broader Impacts

Our proposed method, which enables novel view synthesis and attribute editing of faces from a single image, holds the potential to significantly impact various fields such as film, gaming, augmented reality (AR), and virtual reality (VR). However, it also raises concerns regarding privacy and ethics, particularly the risk of generating "deep fakes". We emphasize the necessity of implementing robust safeguards to ensure the responsible and ethical application of this technology, thereby minimizing the risk of misuse.

## D  Additional Qualitative Results

**Additional Qualitative Evaluation.** We provide more visual comparisons between our WarpGAN and several state-of-the-art methods in Fig. 7. In addition, since we utilize multi-view images synthesized by WarpGAN to assist 3D GAN inversion optimization for editing, we also include comparisons with this optimization-based method (WarpGAN-Opt). We can see that, due to the limitations of the low bit-rate latent code, WarpGAN-Opt loses some detail compared with WarpGAN. However, by leveraging the high-quality novel view images synthesized by WarpGAN, WarpGAN-Opt achieves higher fidelity and realism in novel view synthesis than other optimization-based methods. From the figure, it can be observed that our method outperforms Dual Encoder [4]. However, since our method relies on the visible regions of the input image in the novel view to inpaint occluded regions, our method degrades to a typical encoder-based 3D GAN inversion when the view change is large and the visible region is small. In contrast, Dual Encoder focuses on high-fidelity 3D head reconstruction and thus offers greater flexibility in terms of view changes.

**Additional Attribute Editing Results.** To more comprehensively demonstrate the capability of our method in image attribute editing, we provide additional attribute editing results in Fig. 8. Specifically,

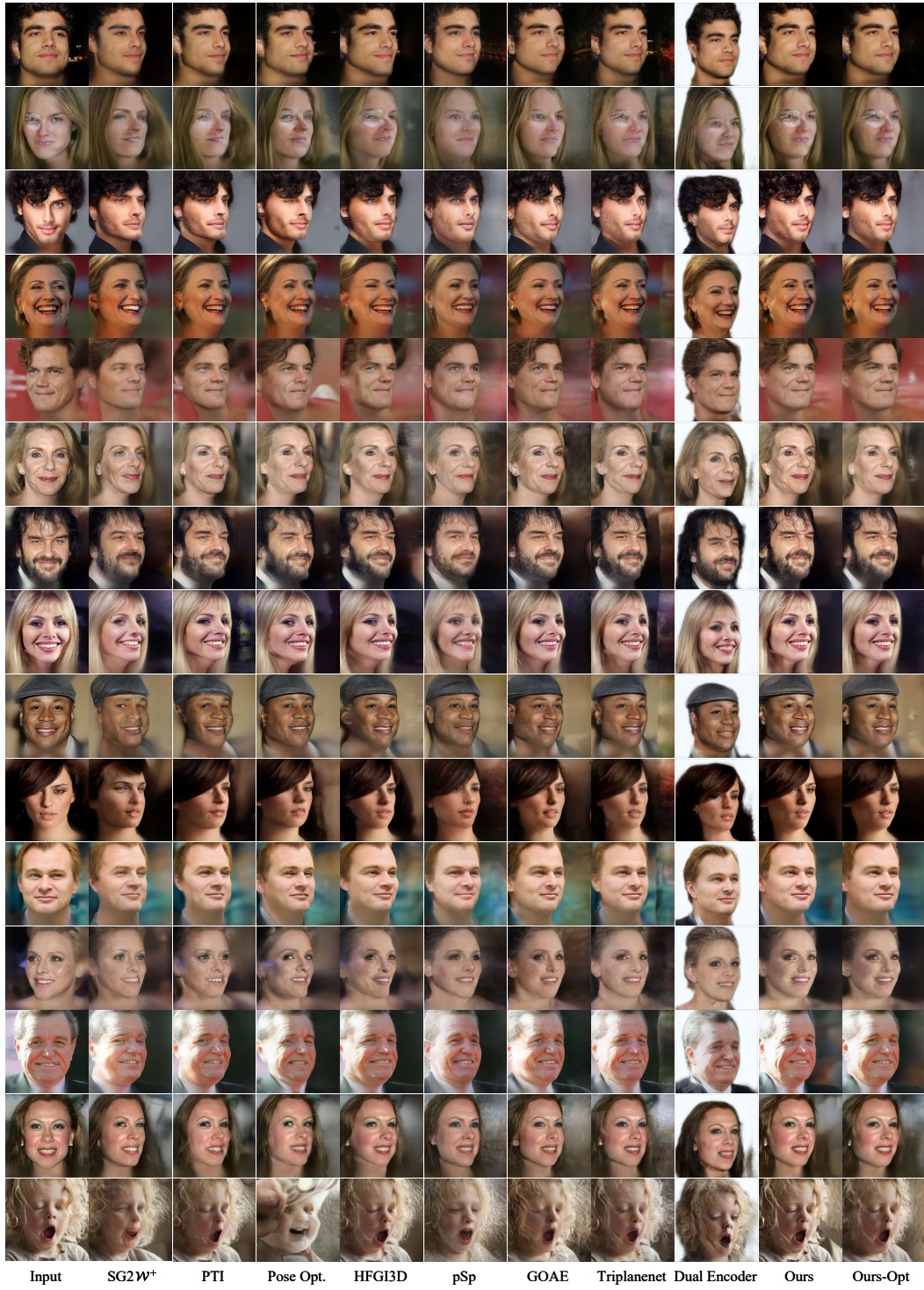

| Input | SG2$\mathcal{W}^+$ | PTI | Pose Opt. | HFGI3D | pSp | GOAE | Triplanenet | Dual Encoder | Ours | Ours-Opt |

Figure 7: **Qualitative comparisons** between our WarpGAN and several state-of-the-art methods.

we employ InterFaceGAN [36] for editing the "Anger", "Old", and "Young" attributes, and utilize the text-guided semantic editing method StyleCLIP [31] for editing the "Elsa" and "Surprised" attributes.

**Reference-Based Style Editing.** In our WarpGAN, the latent code plays a crucial role in controlling the inpainting process of SVINet. To more explicitly analyze the influence of the latent code, we

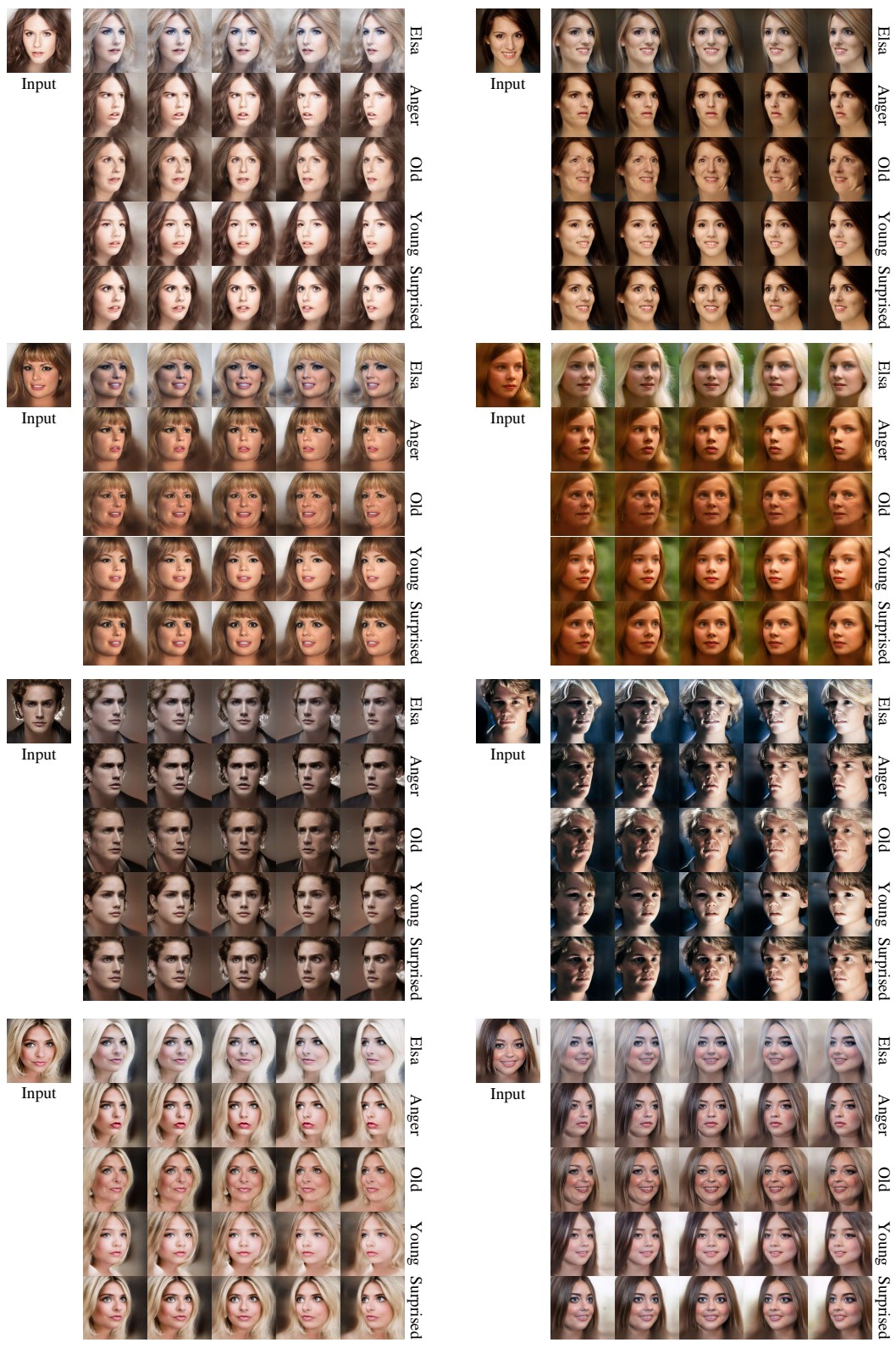

Figure 8: **Image attribute editing results** obtained by our method. The edited attributes include "Elsa", "Anger", "Old", "Young", and "Surprised".

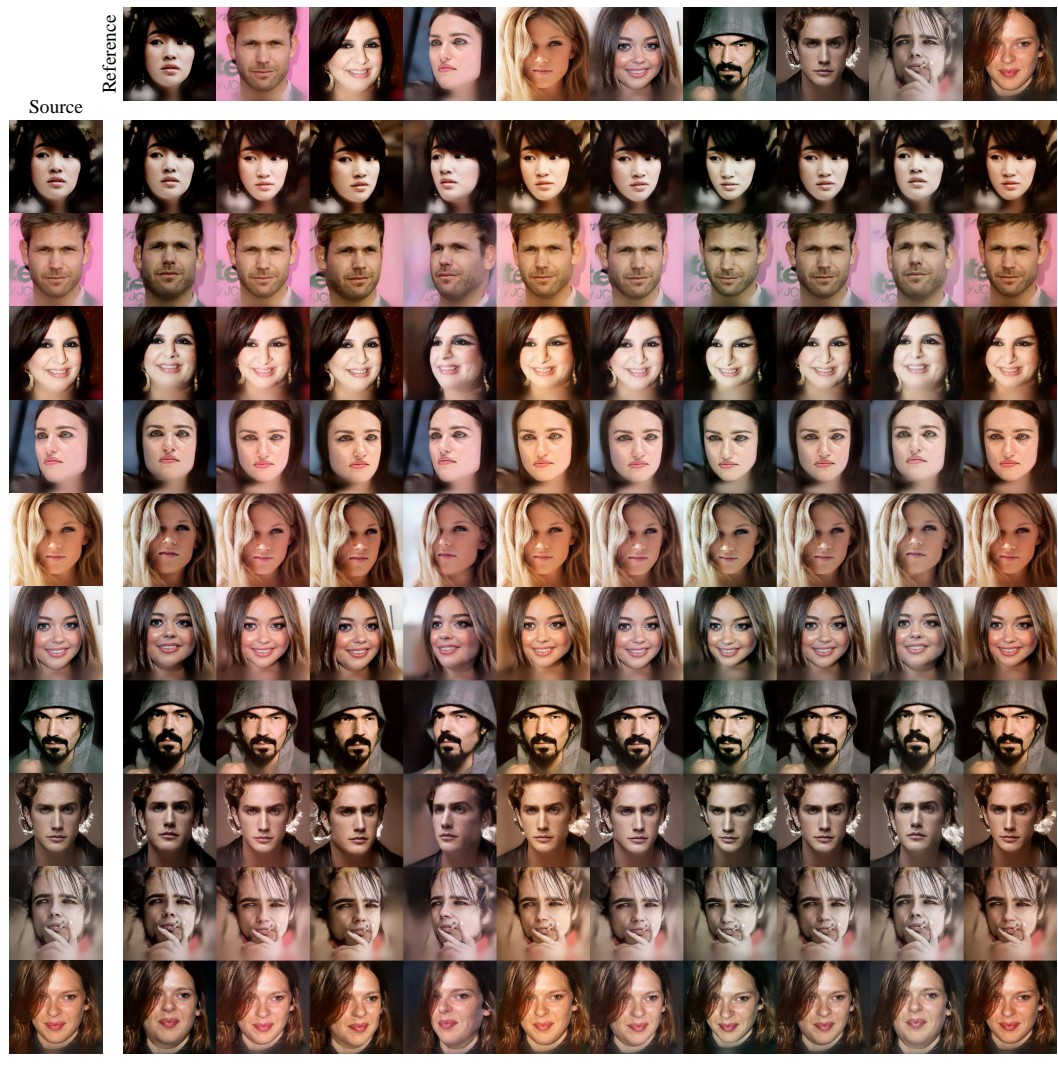

Figure 9: **Reference-based style editing.** Each row represents the editing results of the same source image corresponding to different reference images, where the source and reference images are identical along the diagonal.

perform experiments by replacing the latent code of the input image during the inpainting process. Specifically, for the source image $\mathbf{I}_s$ with the camera pose $c_s$ and the latent code $w_s^+$, we replace them with the camera pose $c_r$ and the latent code $w_r^+$ of the reference image $\mathbf{I}_r$ during inpainting, thereby achieving simultaneous editing of view and style. The results are given in Fig. 9.

For our SVINet, $w^+$ modulates the convolutions in both $N_I$ and $N_D$, where $N_I$ processes feature maps at a resolution of $64 \times 64$, and $N_D$ processes feature maps at resolutions ranging from $64 \times 64$ to $512 \times 512$. According to the characteristics of StyleGAN [20, 21], the latent code corresponding to feature maps at resolutions of $64 \times 64$ and above primarily controls the detailed features of the image, such as the color scheme and microstructure. From Fig. 9, we observe that the main changes are in the skin tone and hair color of the face.

**Qualitative Evaluation in the Cat Domain.** To further validate the generalization capability of our method, we evaluate it in the cat domain. Specifically, we use the AFHQ-CAT dataset [9] for training and evaluation. Following e4e [38], we use a ResNet50 network [16] trained with MOCOv2 [7] instead of the pre-trained ArcFace network [12] to compute the identity loss in the non-facial domains during training. As shown in Fig. 10, our method can generalize well to the cat domain and perform novel view synthesis as well as attribute editing.

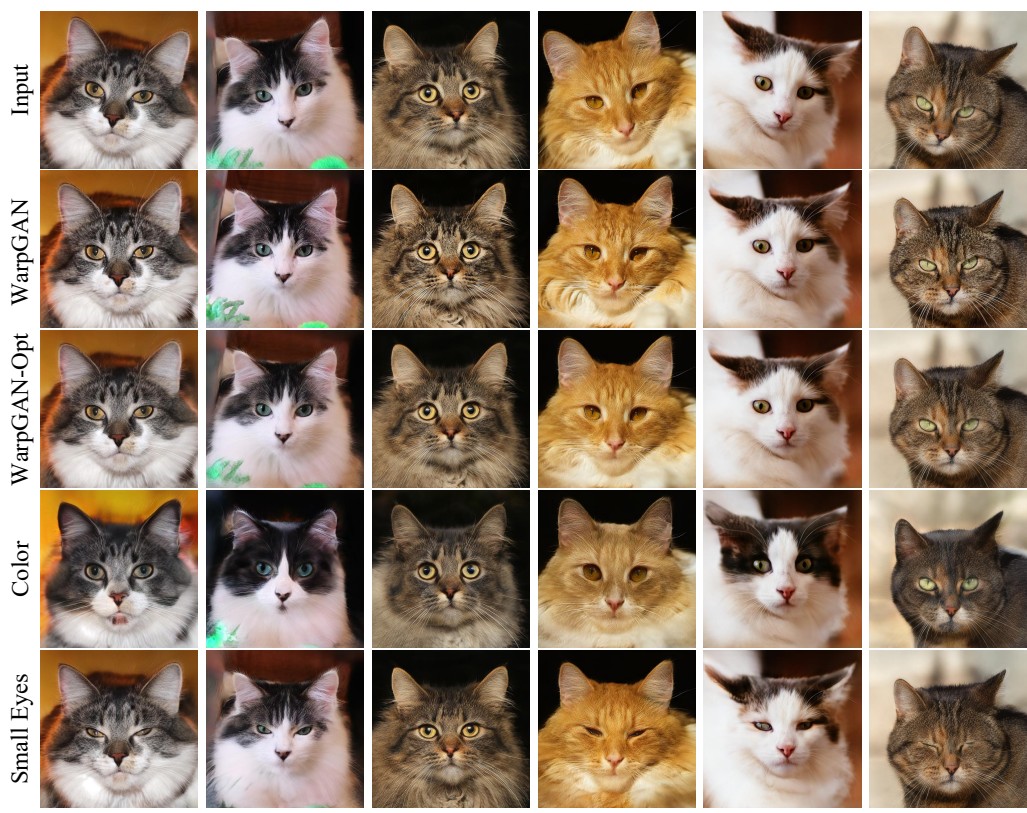

Figure 10: **Novel view synthesis and attribute editing** on cat faces by our method. We visualize the novel view synthesis results of WarGAN and WarpGAN-Opt, as well as the editing results of the attributes "Color" and "Small Eyes".

