# OpenReview forum: "WarpGAN: Warping-Guided 3D GAN Inversion with Style-Based Novel View Inpainting"
_NeurIPS.cc/2025/Conference — NeurIPS 2025 poster_

### Official Review · Reviewer_bp2G · 2025-07-01

**Clarity:** 3
**Significance:** 1
**Originality:** 3
**Rating:** 3
**Confidence:** 4

**Summary:**

This paper presents a method for novel view synthesis of face images, via inversion of an existing 3D-aware GAN (EG3D). It trains an encoder to map from input images to the GAN latent w in order to obtain a depth-map (by decoding and re-rendering); then warps the original image according to the depth map and target pose. Occluded regions are filled in by a custom two-stage in-painting process. The method is evaluated on CelebA-HQ and MEAD, where it shows stronger performance than existing 3D GAN inversion methods.

**Questions:**

Why are both inpainting stages needed? and how are their results combined? In particular, the first inpainting using re-rendered pixels from the inversion (described in eq 2) should result in a complete image; what then is left for the SVINet part to do? It seems the second inpainting is not really inpainting but doing an image-to-image tidying-up of the whole image? What would happen if the second stage were used without the first?

Does the feature-mirroring stage described in sec 3.3.1 presuppose that that the input image is posed symmetrically? What if this is not the case, e.g. a half-profile view is given as input?

What exactly is Figure 3 showing? In particular, it claims to show NVS on CelebA-HQ, but the "input" column appears to contain synthetic images (see the blurring outside of the face region). If these images were in fact generated, this should be stated clearly (since it makes the inversion task easier, as they then definitely lie in the latent manifold of the GAN whereas real images even from the same dataset may not).

**Ethical Concerns:**

["NO or VERY MINOR ethics concerns only"]

**Final Justification:**

Following the rebuttal/discussion, I remain rather unenthused about this paper. The authors have addressed several of the points in my original review adequately, for which I am grateful; however the work still feels dated and not so relevant given modern diffusion approaches to single view reconstruction and novel view synthesis that also work well for broader domains that just the human faces shown here. I therefore keep a negative opinion -- I do not feel it is of sufficient interest to the neurips community. Nonetheless I agree the work is technically sound, and so may rating is only borderline negative.

**Limitations:**

Yes – although limitations is rather brief

**Quality:**

3

**Strengths And Weaknesses:**

## Strengths

The proposed approach is interesting and novel. Explicitly warping the source image ensures that details can be preserved even if they are challenging to encode and decode directly in the GAN's latent space, while still making use of its prior knowledge for depth estimation and occluded regions.

The inpainting stage is itself somewhat novel – it uses a stylegan-inspired architecture that also leverages facial symmetry in feature extraction.

Evaluation covers two large-scale datasets (CelebA-HQ and MEAD); for CelebA-HQ, identity preservation is explicitly measured, as well as realism via FID; for MEAD, reconstruction is also measured directly via LPIPS.

Quantitative results on both datasets are significantly stronger than the chosen baselines according to all metrics. The method is also faster than most of the baselines thanks to being feed-forward (encoder-based) rather than relying on iterative optimization.

The selected qualitative results also appear better than all the baselines, with rather fine details from the source image successfully transferred to the target image.

There is a fairly detailed ablation study, measuring the benefit of various design decisions and innovations.

## Weaknesses

The evaluation only compares against other 3D GAN inversion methods. The comparisons should also include diffusion-based NVS methods (either 2D-only, or 3D-aware) in order to place the method properly in context.

There are no measures of spread (confidence intervals or standard deviations) on the experimental results; this makes it difficult to know if the results are statistically significant. In particular, on certain metrics the gains over the baselines are rather small.

There is no comparison of the full pipeline but with just the warping / rerendering stage replaced by simply using the rerendered inverted result. This comparison is somewhat important, since it justifies the key selling point of the paper, i.e. use of warping, in isolation from confounding factors such as how well the specific inpainting method works.

There is no comparison of the custom SVINet inpainting network against a modern diffusion image-to-image model.

The topic feels somewhat dated – while there is still a strand of literature pursuing inversion of EG3D and similar models as a route to novel view synthesis, the community has largely moved on to diffusion-based methods. This does not in any way affect the technical validity and rigor of the paper; however it does likely limit this work's interest to the community. The restriction to single-view datasets is rather artificial given multi-view datasets of heads are now available; moreover diffusion-based methods such as RenderDiffusion train in EG3D's single-image setting anyway.

The method's applicability is limited to tightly-cropped human heads. Not even EG3D's cats are included, though presumably these would also work in the same pipeline given they are symmetrical too.

---

> ### Author Rebuttal · Authors · 2025-07-31
>
> >**Q1**: Comparison with Diffusion.
>
> **R1**: Our work is a study of 3D GAN inversion, so we primarily compare our method with other 3D GAN inversion-based approaches, as done in SOTA 3D GAN inversion methods (e.g., Triplanenet (WACV'24), In-N-Out (CVPR'24), Dual Encoder (NeurIPS'24)).
> Following your comments, we have also added the performance comparison with the diffusion-based method, DiffPortrait3D [1], on the CelebA-HQ dataset.
> The FID value is 19.26 and the ID value is 0.7964 for DiffPortrait3D, while our WarpGAN achieves an FID value of 19.12 and an ID value of 0.7882. **The metrics indicate that our method can achieve comparable performance to diffusion-based methods on the real human face domain.**
>
> Moreover, due to the uncertainty in the generation of 2D diffusion models and their lack of an explicit 3D representation, multi-view inconsistency is a common issue.
> To address this problem, DiffPortrait3D proposes a view-consistency module that processes multiple novel-view images simultaneously to improve the consistency of the generated novel-view images. However, **such a way requires a significant amount of GPU memory.**
> The authors of DiffPortrait3D set the number of novel-view images to 8 in implementation, which results in the need for inference on at least a single A100.
> Due to the limitation on the number of images processed in parallel, **flickering artifacts** can still be observed in the multi-view rendering animations provided in the DiffPortrait3D demo.
> In contrast, our WarpGAN, benefiting from the multi-view consistency prior of 3D GANs as well as the symmetry prior and latent code from the 3D-aware GAN that we introduce, can easily generate **multi-view consistent rendering videos.** We will provide a demo alongside our open-source code.
>
> It is worth noting that our method can train and infer on consumer-grade GPUs (such as the 3090 and 4090), with inference times on a single 4090 GPU **within milliseconds.** In contrast, DiffPortrait3D requires training on multiple professional-grade GPUs and inference on a single professional-grade GPU, with inference times exceeding **1 minute** on a single A100 GPU.
>
> ---
>
> >**Q2**: Measures of Spread.
>
> **R2**: Statistical significance is indeed crucial for reliable scientific conclusions.
> However, the calculation of confidence intervals or standard deviations is applied mainly to prediction tasks such as object detection and classification, where the ground truth is available.
> In contrast, in the generative domain, there is usually no ground truth of the generated images. The goal is to synthesize new images that conform to certain data distributions, rather than making a decision on a single input. As a result, **most studies in the generative domain do not analyze statistical significance.**
>
> ---
>
> >**Q3**: Small Gains.
>
> **R3**: Although our method shows only minor improvements over optimization-based baselines in certain metrics, our inference speed is significantly faster (over **500 times faster**), demonstrating the efficiency of our method.
>
> ---
>
> >**Q4**: Comparison of the Full Pipeline.
>
> **R4**: Our method consists of two steps: warping and inpainting. We need to warp the input image to the target view first before using SVINet for inpainting. Therefore, **it is NOT feasible to conduct ablation studies separately for these two steps.**
> Regarding the "rerendering" you mentioned, if you are referring to WarpGAN-Opt used in editing operations, we provide extensive visual comparisons and analyses in Fig. 4 of the Appendix.
>
> ---
>
> >**Q5**: Comparison Between SVINet and Diffusion Models.
>
> **R5**: We experimented with the diffusion-based RePaint [2] to inpaint our warped images. Unfortunately, since the holes in our warped images are often large and complex (as shown in Fig. 2), **RePaint, which was not trained in our scenario, struggles to generalize to our task and frequently fails to inpaint the holes.** We also provide metrics on CelebA-HQ, with an FID of **60.79** and an ID score of **0.7390**, indicating poor performance. Additionally, it takes about **5 minutes** per image, which is much slower than our SVINet. However, we believe that modifying diffusion-based methods to replace our SVINet is a promising research direction, and we plan to explore this in the future.
>
> ---
>
> >**Q6**: Outdated GAN.
>
> **R6**: Due to the powerful generative capabilities of diffusion models, which overcome issues such as domain limitations faced by GANs, an increasing number of works in the generative domain adopt diffusion-based methods.
>
> However, **thanks to the strong priors for human heads embedded in 2D/3D/4D GANs, they are capable of generating human heads with rich details.** Although the low-dimensional latent space of GANs imposes domain limitations, it also endows GANs with rich semantic information and powerful disentanglement capabilities within the latent space, enabling finer-grained control over human heads. Given the limitations of diffusion models, such as slower inference speeds and higher computational resource requirements, a significant number of researchers continue to explore GAN-based methods.
>
> Additionally, in the generative domain, diffusion models do not completely outperform GAN-based methods. Both GANs and diffusion models have their own strengths and weaknesses. **Given the powerful generative capabilities of GANs and the strong generalization abilities of diffusion models, many researchers have attempted to combine the two to achieve complementarity.** Here are a few examples:
> 1) $\mathcal{W}_+$ Adapter [3] utilizes StyleGAN (2D GAN) to enhance the disentanglement and facial information retention of diffusion models.
> 2) DiffPortrait3D [1] leverages EG3D (3D GAN) to control portrait novel-view synthesis and employs PanoHead (3D GAN) to generate multi-view images that assist in the training of the diffusion model.
> 3) AvatarArtist [4] uses Next3D (4D GAN) to bridge 2D images and triplanes, and employs a robust 2D diffusion prior to handle diverse data distributions.
>
> Therefore, **we believe that research on GANs remains highly meaningful.** Combining the strengths of GANs and diffusion models to exploit the best of both worlds is a promising research direction. We will also explore integrating diffusion models into our framework in the future.
>
> ---
>
> >**Q7**: Artificial Single-View Datasets.
>
> **R7**: Although multi-view datasets of human heads exist, the requirements for capturing multi-view human heads are quite stringent. Tasks based on single-view images are more aligned with real-world scenarios. Therefore, **research on single-view images is meaningful and has attracted considerable attention [1][4].**
>
> ---
>
> >**Q8**: Comparison with RenderDiffusion.
>
> **R8**: The denoising process of RenderDiffusion resembles EG3D, mapping latent space to 2D images, while its diffusing process parallels 3D GAN inversion, mapping images to latent space. Both learn 3D-consistent representations from 2D images. RenderDiffusion also performs single-image 3D reconstruction. **Although unreleased code prevents direct comparison, the facial reconstruction examples in its paper appear far inferior to our method.**
>
> ---
>
> >**Q9**: Limited to Tightly-Cropped Human Heads.
>
> **R9**: Our method is primarily applied to the task of single-view image reconstruction of human faces. **Due to the limitations of the pre-trained EG3D, our model requires aligned and cropped images as input and can only work within the pre-trained domain of EG3D (e.g., human faces and cats).**
> Under this setup, we are able to achieve high-quality novel-view synthesis of human faces.
> Additionally, we can provide the experiments on cats in the Appendix.
>
> ---
>
> >**Q10**: Two-Stage Inpainting.
>
> **R10**: Since warped images often contain large areas of holes, we first use 3D GAN inversion to roughly fill in the occlusion regions (as shown in Fig. 2). Although the fidelity of the results from 3D GAN inversion is low (corresponding to entry A in Table 2), it can outline the facial contours in the occlusion regions, thereby alleviating the inpainting burden on SVINet. **Given that warped images are somewhat distorted and the initial filling is coarse (corresponding to entry B in Table 2), it is necessary to refine the inpainting in the second stage using SVINet.**
> To clearly demonstrate the role of the first stage, we conducted an ablation study by directly feeding the warped image into SVINet for inpainting. The resulting FID was **19.98** and the ID score was **0.7876**, indicating that the initial coarse filling in the first stage is indeed effective.
>
> ---
>
> >**Q11**: Symmetric Prior.
>
> **R11**: The facial symmetry prior we employ **does NOT assume that the input image is symmetric, but rather that the face structure itself is symmetric.** By incorporating this prior, we extract information from the visible part of the face to inpaint the face on the other side of the occlusion region. As can be seen from the visualizations we provide, **none of our input images are symmetric.**
>
> ---
>
> >**Q12**: Blurring Outside of the Face Region.
>
> **R12**: The input images are real images from CelebA-HQ, but they need to be preprocessed with an alignment and cropping script. Since using the EG3D official preprocessing script to process CelebA-HQ images can result in black borders (as shown in the Triplanenet (WACV'24) paper), **we follow the same preprocessing method of HFGI3D.** Specifically, we first pad the image edges using the "reflect" mode, then perform blurring and other smoothing operations, and finally conduct alignment and cropping.
>
> ---
>
> **References**:
>
> [1] DiffPortrait3D: Controllable Diffusion for Zero-Shot Portrait View Synthesis. CVPR 2024.
>
> [2] RePaint: Inpainting using Denoising Diffusion Probabilistic Models. CVPR 2022.
>
> [3] When StyleGAN Meets Stable Diffusion: a $\mathcal{W}_+$ Adapter for Personalized Image Generation. CVPR 2024.
>
> [4] AvatarArtist: Open-Domain 4D Avatarization. CVPR 2025.

---

> > ### Comment · Reviewer_bp2G · 2025-08-07
> >
> > Thanks for the detailed response; this addresses most concerns adequately. It is reassuring to see the method is comparable with off-the-shelf diffusion based methods.
> >
> > Regarding Q2, the fact that many papers follow poor evaluation protocols does not justify this as good practice. One can (and should) still compute variances / error bars over random seeds, even for generative metrics. Whether a task is generative or discriminative is not relevant.
> >
> > Regarding the 2nd inpainting stage, is this parametrically guaranteed to be multi-view consistent? If not (as it appears from the current description) this seems to weaken the discussion here about multi-view consistency? While it clearly works fairly well in practice, there is a distinction between encouraging consistency vs guaranteeing it, which it would be good to clarify.
> >
> > Regarding Q10 and Q4, my concern here is that there are two innovations presented, but it is difficult to untangle the benefit of each. While you obviously cannot run only half a model, what I feel is missing is a substitution of each part with existing state-of-the-art for each stage. It would be helpful if you could explain why this was not feasible.

---

> > > ### Author Response · Authors · 2025-08-08
> > >
> > > We sincerely appreciate your reply and the helpful suggestions you have provided.
> > >
> > > >**Q1**: Measures of Spread.
> > >
> > > **R1**: We agree with your comments. We understand your emphasis on statistical significance analysis and appreciate your feedback.
> > > For any task, it is indeed feasible to calculate variances by changing the random seed.
> > >
> > > In the generative domain, the intuitive perception of the quality of generated images may be more convincing than quantitative metrics. Therefore, in addition to quantitative analysis, qualitative analysis is also an essential part. We can verify the effectiveness of our method by providing a large number of generated images.
> > > Moreover, the deviations in quantitative metrics caused by the random seed can be minor or even imperceptible from a qualitative perspective. To validate this, we performed additional experiments to change the random seed and calculate the average metric as well as the variance.
> > > Specifically, we set up 5 random seeds and conducted experiments on our method as well as Triplanenet and GOAE on CelebA-HQ. We then calculate the mean and variance of the FID and ID scores for the generated images. The results are given as follows:
> > >
> > > | Method | FID $\downarrow$ | ID $\uparrow$ |
> > > | :----- | :----: | :----: |
> > > | GOAE | 35.48 $\pm$ 0.007293 | 0.7493 $\pm$ 2.830e-07 |
> > > | Triplanenet | 32.60 $\pm$ 0.004040 | 0.7707 $\pm$ 7.805e-09 |
> > > | Ours | 19.11 $\pm$ 0.001967 | 0.7884 $\pm$ 9.516e-08 |
> > >
> > > The results validate the rationality of existing methods. Following your comments, we will give more statistical significance analysis in the supplementary materials.
> > >
> > > ---
> > >
> > > >**Q2**: Discussion on Multi-View Consistency.
> > >
> > > **R2**: Synthesizing multi-view images from a single image is inherently an **ill-posed problem**, and **existing methods can NOT guarantee absolute multi-view consistency** for the generated novel-view images. We discuss our method to address this issue.
> > >
> > > When SVINet fills in the warped image, the model inputs include the warped image, the novel view image generated by 3D GAN inversion, the latent code, and the symmetry prior. Ideally, these inputs are **multi-view consistent** (as the warped image is derived from the input image through warping, while the result of 3D GAN inversion benefits from the 3D-aware GAN. Moreover, the latent code of the same human face from different viewpoints is consistent).
> > > Therefore, ideally, the output of our model would also be multi-view consistent.
> > >
> > > However, we cannot fully guarantee that this situation holds completely.
> > > 1) **Due to the errors in the depth map synthesized by 3D GAN inversion, the warped image may have certain distortions at different views.** SVINet needs to make some corrections to it. Therefore, the warped image cannot be guaranteed to fully conform to the ideal situation.
> > > 2) Since SVINet is a deep learning network with a complex structure, there is also **some uncertainty in the generated results**.
> > >
> > > Therefore, we cannot confidently assert that we can fully guarantee multi-view consistency.
> > >
> > > Nevertheless, through extensive quantitative and qualitative analyses, it can be seen that compared with existing SOTA methods based on 3D GAN inversion, **our method achieves the best results in addressing this issue**. Moreover, compared with methods based on diffusion models that have greater randomness, **our method also shows a clear advantage in solving this problem**.
> > >
> > > It can also be seen from the multi-view rendering videos generated from a single image by our method that our method can maintain good multi-view consistency. We will provide some demos alongside our open-source code.

---

> > > > ### Author Response · Authors · 2025-08-08
> > > >
> > > > >**Q3**: Comparison of the Full Pipeline.
> > > >
> > > > **R3**: Thank you for restating the question. We have analyzed the feasibility of replacing our two steps with existing SOTA methods as you mentioned:
> > > > 1) **Replacing our less-performant 3D GAN inversion with SOTA 3D GAN inversion**: We provided a similar response when answering **Reviewer kyEW**'s Q2, and we reiterate it here for your question. Recently, there have been many works on 3D GAN inversion, including optimization-based and encoder-based methods. Optimization-based methods are time-consuming, while encoder-based methods have lower fidelity. To improve the fidelity of encoder-based methods, some methods augment the high-dimensional $\mathcal{F}$ space. However, GOAE (ICCV'23) mentioned that this method can lead to distortion in the occlusion region. Since the primary role of 3D GAN inversion in our framework is to generate depth maps and to preliminarily fill in the occlusion regions in the warped image, using a method that augments the $\mathcal{F}$ space with higher fidelity would actually degrade the preliminary filling effect. Therefore, **our 3D GAN inversion only uses the most basic technique, which is to project the image into the low-dimensional $\mathcal{W}$ space of EG3D**.
> > > > 2) **Replacing our SVINet with a SOTA inpainting network**: As we mentioned in our **R5**, we attempted to use the powerful diffusion model-based method RePaint for inpainting, but the results were not satisfactory. The reason is that **the holes in our warped images are often large and complex** (as shown in Fig. 2, including point-like, line-like, and block-like holes), and inpainting networks (which are not trained in our specific scenario) are difficult to generalize directly to our task, often failing to fill in the holes. Moreover, when we use existing image inpainting methods, the quality of the warped images to be inpainted often cannot be guaranteed, with some distortions present.

---

> > > > > ### Comment · Reviewer_bp2G · 2025-08-08
> > > > >
> > > > > I see; thanks for the clarification on these points.

---

> > > > > > ### Author Response · Authors · 2025-08-08
> > > > > >
> > > > > > We are deeply grateful to all the reviewers for their valuable feedback. We appreciate all the reviewers' acknowledgment that our responses resolved their concerns. We sincerely welcome the reviewers' reconsideration of the rating. We make a commitment that all the raised concerns will be properly addressed in the final version.

---

### Official Review · Reviewer_rVfV · 2025-07-02

**Clarity:** 4
**Significance:** 3
**Originality:** 4
**Rating:** 4
**Confidence:** 4

**Summary:**

This paper tackles the problem of low-quality novel view synthesis in single-image 3D GAN inversion, focusing on improving occluded regions that often suffer from missing details and a lack of realism. The authors propose WarpGAN, a novel method that integrates the classical "warping-and-inpainting" strategy into the 3D GAN inversion pipeline. Specifically, the method first warps the input image to a new viewpoint using a depth map obtained from the 3D GAN inversion, thereby preserving high-frequency details in visible areas. Then, it employs a style-based inpainting network, SVINet, which leverages facial symmetry priors and the latent code from the 3D GAN to realistically fill in the occluded regions while maintaining multi-view consistency. To overcome the lack of ground-truth data for real images, the authors introduce a "re-warping" strategy. Quantitative and qualitative experimental results demonstrate that the proposed method outperforms existing state-of-the-art approaches.

**Questions:**

Please see the raised weaknesses.

**Ethical Concerns:**

["NO or VERY MINOR ethics concerns only"]

**Final Justification:**

I thank the authors' rebuttal, which addresses part of my concerns regarding the novelty of the proposed method. My comments about extreme cases still hold but I agree it is a challenge for the community. Therefore, I keep my positive rating and have no objection to accept.

**Limitations:**

Yes.

**Quality:**

4

**Strengths And Weaknesses:**

Strengths
1. Clear Motivation: The over-reliance on the 3D GAN's generative prior to fill occluded regions leads to poor generation quality. The Integration of the "Warping-and-Inpainting" framework is reasonable.
2. Innovative SVINet Network: The symmetry prior and style-based modulation used in the SVINet and its training strategy is justified and convincing.
3. Good results: Current experiments demonstrate the effectiveness of the proposed method.


Weaknesses
1. The core idea of integrating a "warping-and-inpainting" strategy with 3D GAN inversion is not new and has been previously explored by methods like HFGI3D. The authors argue that their key contribution lies in replacing the optimization-based inpainting of HFGI3D with a dedicated feed-forward network (SVINet), which is incremental. Therefore, the paper could be strengthened by more clearly articulating and demonstrating why its proposed feed-forward inpainting network is superior to a well-formulated optimization-based alternative, perhaps by highlighting scenarios where optimization-based methods fundamentally fail and only a dedicated, trained network can succeed.
2. The method's generalization ability to more "in-the-wild" scenarios—such as extreme poses, exaggerated expressions, complex occluders, or non-face objects—remains to be validated. For example, the symmetry prior might not hold when a hand occludes the face.


Minor:
1. Line 216 “Datesets” typo

---

> ### Author Rebuttal · Authors · 2025-07-31
>
> >**Q1**: Incremental Innovation over HFGI3D.
>
> **R1**: WarpGAN is NOT an incremental innovation over HFGI3D.
> 1) **HFGI3D is NOT a combination of the "warping-and-inpainting" strategy and the 3D GAN inversion.** The core idea of HFGI3D is to use warping operations to synthesize pseudo-multi-view images, where the occlusion regions of the warped images are simply filled by using the results of PTI. However, such a filling operation is coarse. Since PTI is optimized by using a single-view image and has low fidelity, it leaves obvious discontinuities when combined with the warped image (similar to the coarse result $\hat{\mathbf{I}}_{novel}^{\text{initial}}$ from 3D GAN inversion, as shown in Fig. 2 of our paper). We believe that such a way is merely a simple fill rather than true inpainting, and the synthesized pseudo novel-view images cannot be regarded as ideal results. Hence, they still require a final optimization. In contrast, our SVINet can truly inpaint the warped image and directly generate natural and plausible occlusion regions.
> 2) **Our method is NOT a simple replacement of optimization-based approaches with feed-forward methods.** The purpose of synthesizing pseudo-multi-view images in HFGI3D is to assist optimization.
> However, since PTI optimizes solely based on a single-view image, it is prone to generating novel-view images with poor geometry quality. Using PTI to fill in the warped image results in pseudo-multi-view images with inferior geometry quality.
> Moreover, inevitable errors in the depth map can make warped images unreliable under large view changes, thereby affecting the quality of pseudo-multi-view images. Consequently, low-quality pseudo-multi-view images can have a negative impact on the final optimization. In comparison, the occlusion regions synthesized by our SVINet are of higher quality than those filled by PTI and have the ability to correct unreliable warped images to some extent (as shown in Fig. 5(b)).
> As shown in Fig. 5(c), the geometry quality of the novel-view images generated by WarpGAN-Opt is better than that of the results from HFGI3D. This indicates that the novel-view images synthesized by our WarpGAN have better quality than the pseudo-multi-view images generated by HFGI3D.
>
> ---
>
> >**Q2**: Unvalidated Generalizability.
>
> **R2**: The extreme cases you mentioned, such as extreme poses, exaggerated expressions, complex occluders, or non-face objects, are indeed challenging for methods based on 3D GAN inversion. Existing methods generally struggle to perform well under these circumstances.
> We discuss the generalizability of our method to these extreme cases here in principle and will supplement the performance of our method under these extreme conditions in the Appendix.
>
> For **extreme poses**, although our method can generate high-quality novel-view images, there are certain limitations in the range of pose changes. We mentioned the limitations of our method in the Conclusion section of the paper. Due to inevitable errors in the depth map, the quality of the images generated by our WarpGAN will decrease when the camera pose changes significantly. More seriously, when the camera pose changes significantly, leaving no visible area in the novel view, our method will degrade to a typical encoder-based 3D GAN inversion (as mentioned in lines 76-79 of the Appendix).
>
> For **exaggerated expressions and complex occluders**, which can be collectively referred to as out-of-domain regions in the GAN field, existing GAN inversion methods, limited by the low-dimensional latent space of GANs, struggle to reconstruct these out-of-domain regions effectively. However, the visible regions of the novel-view images synthesized by our method come from the original image, and the occlusion regions come from SVINet, which can bypass the information loss caused by the low-dimensional latent space and thus better preserve these out-of-domain regions.
>
> For **non-face objects**, EG3D provides a pre-trained model for cats, and we will evaluate our method on cats in the Appendix. Faces are more complex than cats. Thus, in principle, our method should be able to generalize well to novel-view synthesis of cats.
>
> Regarding **symmetry-aware feature extraction**, as shown in Eq. (4), we do not directly add the features of the input image and its mirrored image. Instead, we use featurewise linear modulation (FiLM) to refine the features, which can be considered as a weighted fusion. When the input face image is significantly asymmetric, the symmetric weight will decrease. Similarly, Triplanenet (WACV'24), when using symmetry priors for loss calculation, introduces a per-pixel Gaussian density with a pixelwise uncertainty map that assigns lower confidence to the region in the mirrored image where the symmetry assumption fails. Therefore, our explanation is justified in theory, and we will also supplement our performance when the face prior does not work (such as in the case of ``hand covering face'' that you mentioned) in the Appendix.
>
> ---
>
> >**Q3**: Typo.
>
> **R3**: Thank you for your attention to detail. We will correct the typo in the revision.

---

> > ### Comment · Reviewer_rVfV · 2025-08-06
> >
> > Thank you so much for the rebuttal, which addresses most of my concerns.

---

> > > ### Author Response · Authors · 2025-08-06
> > >
> > > Thank you for your valuable response. We truly appreciate your thorough review and constructive feedback.

---

### Official Review · Reviewer_kyEW · 2025-07-02

**Clarity:** 3
**Significance:** 3
**Originality:** 3
**Rating:** 4
**Confidence:** 4

**Summary:**

This paper proposes a warping-then-inpainting approach to synthesize multi-view images for facial images.  It first employs a EG3D-style GAN to do the 3D  inversion of  a single view facial image, produce its depth map, and then inpaint the warped facial image at the new viewpoint.  The proposed approach can also enable facial editing in the latent feature space.  The experimental results show the improvement of quantitative metrics, such as FID and ID.

**Questions:**

1. Does the row A indicate the result of 3D GAN  inversion?  Since there are recent studies on 3D facial image editing based on 3D GAN  inversion, I am curious why the results of 3D GAN inversion only are much worse than warp-then-inpaint.
2. Since warp-then-inpaint procedure is widely used in diffusion-based methods,  what is the comparison between the proposed approach and these diffusion-based methods?  please also explain why diffusion-based models are not used.

**Ethical Concerns:**

["NO or VERY MINOR ethics concerns only"]

**Final Justification:**

My concerns are addressed in the rebuttal. I maintain my borderline accept score as the final rating.

**Limitations:**

Yes

**Paper Formatting Concerns:**

No concerns

**Quality:**

3

**Strengths And Weaknesses:**

Strengths:
1.  It shows that warp-then-inpaint procedure can improve the quality of the novel view synthesis of facial images. Experimental results are impressive.
2. The ablation study clearly shows the improvements by SVINet.

Weaknesses:
While the experimental results are impressive,  the technical components in the proposed approach are mainly adapted from already-known methods, such as EG3D, symmetry and ID  loss.  It downgrades the technical novelty of this paper.

---

> ### Author Rebuttal · Authors · 2025-07-31
>
> >**Q1**: Novelty.
>
> **R1**: Although the technical components are established techniques, we do not simply stack them. Instead, **we are the first to successfully apply the "warping-and-inpainting" paradigm to 3D GAN inversion, by innovatively designing a warping-back strategy and introducing a symmetry prior along with latent codes from a 3D-aware GAN, greatly alleviating multi-view inconsistency.** Our work clearly showcases the great potential of the "warping-and-inpainting" paradigm in 3D GAN inversion.
> Qualitative and quantitative results confirm the effectiveness of our design.
> We believe that the fast inference time, together with the good performance of the framework, will benefit and ease future research on novel-view synthesis of human faces.
>
> ---
>
> >**Q2**: Poor Performance of 3D GAN Inversion.
>
> **R2**: Entry A in Table 2 reports results obtained directly from the initial novel-view images produced by 3D GAN inversion.
> Recent 3D GAN inversion methods fall into two categories: optimization-based and encoder-based. Optimization methods are time-consuming, whereas encoder-based methods suffer from low fidelity. To boost the fidelity of encoder-based methods, some works complement the high-dimensional $\mathcal{F}$ space, yet GOAE (ICCV'23) shows that such a way introduces distortion in occluded regions.
> Because the primary role of 3D GAN inversion in our method is to produce a depth map and provide an initial fill for occluded regions in the warped image, we adopt the most basic approach (Entry A): simply projecting the image into the low-dimensional $\mathcal{W}$ space of EG3D. **This inevitably incurs substantial information loss, which in turn leads to lower fidelity in the novel-view images.** In contrast, our full model retains the visible regions from the original image and generates occlusions with SVINet, thereby bypassing the information loss inherent to the low-dimensional latent space and achieving much better results.
>
> ---
>
> >**Q3**: Comparison with Diffusion-Based Methods.
>
> **R3**: The "warping-and-inpainting" strategy in diffusion models (such as GenWarp(NeurIPS'24) and LucidDreamer(ICCV'24)) is primarily designed for novel-view synthesis of scenery. In contrast, our work is object-centric, focusing on novel-view synthesis of human faces.
> Moreover, their training datasets and problem settings differ from ours. Hence, **these methods cannot be directly transferred to our task.**
>
> Following the suggestion of **Reviewer bp2G**, we have also added the performance comparison with a representative diffusion-based method, DiffPortrait3D [1], on the CelebA-HQ dataset. The FID value is **19.26** and the ID value is **0.7964** for DiffPortrait3D, while our WarpGAN achieves an FID value of **19.12** and an ID value of **0.7882**.
> **The metrics indicate that our method can achieve performance comparable to diffusion-based methods in the real human face domain.**
>
> Moreover, due to the uncertainty in the generation of 2D diffusion models and their lack of an explicit 3D representation, multi-view inconsistency is a common issue.
> To address this problem, DiffPortrait3D proposes a view-consistency module that processes multiple novel-view images simultaneously to improve the consistency of the generated novel-view images. However, such a way requires a significant amount of GPU memory.
> The authors of DiffPortrait3D set the number of novel-view images to 8 in implementation, which results in the need for inference on at least a single A100.
> Due to the limitation on the number of images processed in parallel, **flickering artifacts** can still be observed in the multi-view rendering animations provided in the DiffPortrait3D demo.
> In contrast, our WarpGAN, benefiting from the multi-view consistency prior of 3D GANs as well as the symmetry prior and latent code from the 3D-aware GAN that we introduce, can easily generate **multi-view consistent rendering videos**. We will provide a demo alongside our open-source code.
>
> It is worth noting that our method can train and infer on consumer-grade GPUs (such as the 3090 and 4090), with inference times on a single 4090 GPU **within milliseconds**. In contrast, DiffPortrait3D requires training on multiple professional-grade GPUs and inference on a single professional-grade GPU, with inference times exceeding **1 minute** on a single A100 GPU.
>
> ---
>
> >**Q4**: No Diffusion Used.
>
> **R4**: We discuss this from three perspectives:
> 1) Our work is a study of 3D GAN inversion, so we primarily consider using GAN-based methods to achieve novel-view synthesis of human faces from a single image, as done in SOTA 3D GAN inversion methods (e.g., Triplanenet (WACV'24), In-N-Out (CVPR'24), Dual Encoder (NeurIPS'24)).
> Diffusion models are trained on massive image corpora that inevitably cover the celebrities present in CelebA-HQ. **Hence, it would be unfair to compare them with 3D GAN-inversion methods.**
> 2) 3D-aware GANs can provide strong facial priors and multi-view-consistent generation, together with a latent space that is semantically rich and highly disentangled. **We thus believe that advancing 3D GAN inversion itself is worthwhile.** Extensive experiments demonstrate that our WarpGAN delivers high-quality novel-view face synthesis, runs faster, and consumes fewer resources than diffusion-based alternatives.
> 3) In future work, we believe that **incorporating diffusion models into our framework** is a promising research direction, such as attempting to use diffusion-based methods to replace SVINet for inpainting occlusion regions.
>
> ---
>
> **References**:
>
> [1] DiffPortrait3D: Controllable Diffusion for Zero-Shot Portrait View Synthesis. CVPR24.

---

> ### Comment · Reviewer_kyEW · 2025-08-06
> **Maintain my score**
>
> After reading the rebuttal, my concerns are basically addressed. I still maintain my viewpoint that the main efforts of this paper are devoted to improving the quality of synthesis results.

---

> > ### Author Response · Authors · 2025-08-06
> >
> > Thank you for your valuable comments and feedback.
> >
> > In our paper, we introduce the “warping-and-inpainting” strategy to 3D GAN inversion for the first time. Our method offers significant advantages in both effectiveness and efficiency. Different from traditional methods that require mapping the input image to the latent space, our method bypasses this step to enhance the quality of synthesizing novel-view images,  inspiring future research for 3D GAN inversion.
> >
> > If you have any further concerns, please feel free to give comments.

---

### Official Review · Reviewer_bFFc · 2025-07-02

**Clarity:** 3
**Significance:** 3
**Originality:** 3
**Rating:** 4
**Confidence:** 4

**Summary:**

This paper presents a novel method, WarpGAN, aimed at addressing the occlusion problem in the task of 3D GAN inversion. The authors first employ an encoder to project the input image into a latent code in the w space, which is then fed into a 3D GAN to synthesize views. A carefully designed warping strategy is subsequently applied to generate novel views along with corresponding synthetic depth maps. Finally, an inpainting network is introduced further to enhance the visual fidelity of the generated views. Experimental results demonstrate the effectiveness of the proposed approach.

**Questions:**

Please refer to the Strengths And Weaknesses section.

**Ethical Concerns:**

["NO or VERY MINOR ethics concerns only"]

**Limitations:**

Yes.

**Paper Formatting Concerns:**

None.

**Quality:**

3

**Strengths And Weaknesses:**

Strengths:

1.	The paper is well-written and easy to follow.

2.	The use of the warping-and-inpainting strategy to handle the occlusion problem is an interesting and effective design choice.

3.	WarpGAN achieves considerable improvements over baseline methods, demonstrating the effectiveness of the proposed approach.

Weaknesses:

1.	While the authors claim to address the occlusion problem, the experimental section lacks a specific analysis of how occlusion affects the performance of baseline methods. For example, does occlusion lead to artifacts in facial regions such as the ears or side profile? It would strengthen the paper if the authors explicitly highlighted the impact of occlusion on visual quality and reconstruction accuracy.

2.	The authors generate synthetic training samples, but as shown in [1], naively sampling from EG3D can produce unstable results. Therefore, more details about the sampling strategy are needed to assess the reliability and quality of the training data.

3.	The organization of the paper could be improved. For instance, Section 3.1 currently focuses heavily on warping. It may be clearer to introduce a dedicated subsection for the warping module and keep the overview section as a concise summary of the entire pipeline.


[1] G-NeRF: Geometry-enhanced Novel View Synthesis from Single-View Images. CVPR 2024.

---

> ### Author Rebuttal · Authors · 2025-07-31
>
> >**Q1**: Impact of Occlusion.
>
> **R1**: We evaluate our method against baseline methods from both quantitative and qualitative perspectives to clearly reveal how occlusion affects each method. Owing to space limitations, the original paper offers only a brief discussion. Here we provide a more complete analysis.
> 1) **Quantitatively**, FID reflects the overall quality of the synthesized novel views, while ID measures the identity preservation and the plausibility of the generated occluded regions to some extent. The LPIPS metric on MEAD further reflects the reconstruction fidelity of novel views and, indirectly, the accuracy of the occluded areas. Across these metrics, our method achieves the best results, suggesting that baselines produce less realistic occlusions.
> 2) **Qualitatively**, extensive visual comparisons are presented in Fig. 3 and Fig. 4 of the main paper, as well as Fig. 4 in the supplementary materials. They demonstrate that for regions unobserved in the input (e.g., ears or side profiles), baselines tend to yield blurry artifacts or unnaturally flattened heads, whereas our WarpGAN generates occluded facial regions that are both more coherent and richer in detail.
>
> ---
>
> >**Q2**: Sampling Strategy for Synthetic Samples.
>
> **R2**: To facilitate training and enhance multi-view consistency, we leverage the powerful generative capacity of EG3D to generate multi-view-consistent synthetic images as additional supervision.
> G-NeRF claims that regions of lower density in the latent space of StyleGAN may be inadequately represented, leading to synthetic images with degraded geometry quality.
> After checking the synthetic images used in our training, we did not observe the fidelity or visual-appeal issues reported by G-NeRF. We attribute this to two key reasons.
> 1) **EG3D checkpoint**:
>    As G-NeRF does not release the code for generating synthetic images, the exact details of their sampling strategy remain unknown. However, the EG3D checkpoint they provide was trained on the standard FFHQ dataset, whereas we use a checkpoint fine-tuned on a rebalanced FFHQ with a more uniform pose distribution. According to the EG3D authors, such a fine-tuned model produces better 3D shapes and better renderings from steep angles than the one trained on the standard FFHQ. We therefore consider this to be a reason for our observations.
> 2) **Sampling ranges**:
>    For latent codes, we sample from a Gaussian distribution and then map to the $\mathcal{W}$ space (as mentioned in the G-NeRF paper). For camera poses, because the FFHQ dataset has a limited pose distribution, sampling over a wider range can introduce artifacts. Therefore, our sampling range primarily focuses around the frontal view of the face: yaw and pitch are both drawn within approximately $\pm 30^{\circ}$ of the frontal view. Poor geometry quality typically appears at extreme angles, so our constrained sampling avoids the issues highlighted by G-NeRF.
>
> ---
>
> >**Q3**: Organization of the Paper.
>
> **R3**: Thank you for your valuable feedback. We agree that introducing warping as a Preliminary at the beginning of the Methodology section will make our Overview more concise and improve the readability of the paper. We will revise our paper according to your suggestions.

---

### Official Review · Reviewer_bJgh · 2025-07-02

**Clarity:** 3
**Significance:** 2
**Originality:** 2
**Rating:** 4
**Confidence:** 4

**Summary:**

The paper tackles the long‑standing problem of poor occluded‑region quality in single‑image 3D‑aware GAN inversion. It combines depth‑based warping (to reuse pixels that are already visible) with a dedicated style‑based 2‑D inpainting network (SVINet) that is conditioned on (1) the latent code produced by an EG3D encoder and (2) a symmetry prior obtained from a horizontally flipped copy of the input face. Training mixes real single‑view images with synthetic multi‑view pairs rendered from EG3D. Quantitative results on CelebA‑HQ and on MEAD show reasonable gains over eight published baselines, and inference is as fast as other feedforward. Ablations support the contribution of each design choice. The authors also demonstrate that the synthesized multi‑view set can be fed to PTI to obtain superior latent‑space editing.

**Questions:**

N/A

**Ethical Concerns:**

["NO or VERY MINOR ethics concerns only"]

**Final Justification:**

I do not have much problems with the paper initially, and would keep my positive score.

**Quality:**

3

**Strengths And Weaknesses:**

Strength:

-- The paper solves a clear problem and is well-motivated. 3D GAN inversion does tend to hallucinate unseen regions, the paper addresses an interesting problem.

-- The warping and inpainting approach is reasonable, since the depth maps come out for free from EG3D, and conditioning SVINet on the latent code definitely improves multi-view consistency.

-- The improvements are significant with limited training budget.


Weakness:

-- This might go back to the old "novelty" problem, but I do feel the novelty of this paper is incremental relative to existing “warp + inpaint” pipelines. The idea has appeared in GenWarp (NeurIPS’24) and LucidDreamer (ICCV’24), and In‑N‑Out (CVPR’24) already mixes triplane editing with an occlusion‑aware decoder. The paper does not position itself clearly against these very closely related works.

-- The evaluation domain is kind of limited, since it is face only.


Overall, the paper offers a practically useful recipe that materially improves facial 3D inversion quality at encoder‑level speed. However, its methodological novelty over other warp‑and‑inpaint works is moderate, the evaluation is narrowly scoped to faces. It is a decent paper, but the contribution may be judged as incremental for NeurIPS.

---

> ### Author Rebuttal · Authors · 2025-07-31
>
> >**Q1**: Novelty.
>
> **R1**: Existing "warping-and-inpainting" methods cannot be directly used in 3D GAN inversion (lines 40-42). **We are the first to successfully apply such a paradigm to 3D GAN inversion. We innovatively design a warping-back strategy and introduce a symmetry prior along with latent codes from a 3D-aware GAN, greatly alleviating multi-view inconsistency.**
> This enables novel-view synthesis of a human face from a single image, achieving performance superior to optimization-based methods with the inference time comparable to encoder-based methods. Hence, our work clearly showcases the great potential of the "warping-and-inpainting" paradigm in 3D GAN inversion.
> We believe that the fast inference time, together with the good performance of the framework, will benefit and ease future research on novel-view synthesis of human faces.
>
> ---
>
> >**Q2**: Comparisons with SOTA Methods.
>
> **R2**: Our method is intrinsically different from GenWarp (NeurIPS'24) and LucidDreamer (ICCV'24), which are specifically designed for **novel-view synthesis of scenery**; in contrast, our work is object-centric, focusing on **novel-view synthesis of human faces.** Moreover, their training datasets and problem settings differ from ours. Hence, these methods cannot be directly transferred to our task.
>
> We provide an extra comparison with In‑N‑Out (CVPR'24) on the CelebA-HQ dataset. The authors of In‑N‑Out evaluated their method on a custom dataset that includes the mask information marking out-of-domain regions. To make a fair comparison with our method and other baselines, we apply In‑N‑Out to the CelebA-HQ dataset, where **it achieves an FID value of 60.41 and an ID value of 0.6744, indicating poor performance.**
> One possible reason is that CelebA-HQ does not have annotations for marking out-of-domain regions. Therefore, during optimization, we use the method (provided by the authors) that does not include the mask in the loss calculation. This may directly cause the additionally optimized tri-plane to fail to reconstruct out-of-domain regions well, leading to a performance drop.
>
> In addition, following the suggestion of **Reviewer bp2G**, we have also added the performance comparison with a representative diffusion-based method, DiffPortrait3D [1], on the CelebA-HQ dataset. The FID value is 19.26 and the ID value is 0.7964 for DiffPortrait3D, while our WarpGAN achieves an FID value of 19.12 and an ID value of 0.7882.
> **The metrics indicate that our method can achieve performance comparable to diffusion-based methods in the real human face domain.**
>
> Moreover, due to the uncertainty in the generation of 2D diffusion models and their lack of an explicit 3D representation, multi-view inconsistency is a common issue.
> To address this problem, DiffPortrait3D proposes a view-consistency module that processes multiple novel-view images simultaneously to improve the consistency of the generated novel-view images. However, such a way requires a significant amount of GPU memory.
> The authors of DiffPortrait3D set the number of novel-view images to 8 in implementation, which results in the need for inference on at least a single A100.
> Due to the limitation on the number of images processed in parallel, **flickering artifacts can still be observed in the multi-view rendering animations provided in the DiffPortrait3D demo**.
> In contrast, our WarpGAN, benefiting from the multi-view consistency prior of 3D GANs as well as the symmetry prior and latent code from the 3D-aware GAN that we introduce, can easily generate **multi-view consistent rendering videos**. We will provide a demo alongside our open-source code.
>
> It is worth noting that our method can train and infer on consumer-grade GPUs (such as the 3090 and 4090), with inference times on a single 4090 GPU **within milliseconds.** In contrast, DiffPortrait3D requires training on multiple professional-grade GPUs and inference on a single professional-grade GPU, with inference times exceeding **1 minute** on a single A100 GPU.
>
>
> ---
>
> >**Q3**: Limited Evaluation Domain.
>
> **R3**: We follow state-of-the-art 3D GAN inversion methods (e.g., Triplanenet (WACV'24), In-N-Out (CVPR'24), Dual Encoder (NeurIPS'24)) to validate our method in the human face domain. We agree with you that evaluating our method in other domains is a worthwhile endeavor. Note that EG3D provides a pre-trained model for cats. Compared with faces, which contain rich details, cat reconstruction is relatively simpler. Therefore, **we believe our method, in theory, can perform comparably well on cats as it does on faces.** We will supplement this evaluation in the Appendix.
>
> ---
>
> **References**:
>
> [1] DiffPortrait3D: Controllable Diffusion for Zero-Shot Portrait View Synthesis. CVPR 2024.

---

> > ### Comment · Reviewer_bJgh · 2025-08-05
> >
> > I do not have many concerns initially and would keep my positive score.

---

> > > ### Author Response · Authors · 2025-08-06
> > >
> > > Thank you for the valuable feedback. We sincerely appreciate your positive comments on our work.

---

### Note · Authors · 2025-08-12

We are grateful to all the reviewers for their valuable comments, which have been very helpful in improving the quality of our paper. Following the discussion, all reviewers who participated mentioned that most of their concerns were addressed. Although one reviewer did not participate in the discussion, we still thank him for the positive rating he gave us and have provided detailed explanations for the questions he raised.

---

### Decision · Program_Chairs · 2025-09-17

**Decision:**

Accept (poster)

**Comment:**

The paper proposes WarpGAN, a method to improve novel view synthesis in single-image 3D GAN inversion by addressing occluded regions through a combination of depth-based warping and a style-based inpainting network (SVINet). SVINet leverages latent codes from EG3D and a facial symmetry prior to enhance realism and consistency in unseen areas. The experiments show consistent improvements over the baselines, and ablations confirm the contribution of each component.

Reviewers liked the clear motivation, strong results with fast inference, and practical relevance for applications like latent editing.

Weaknesses mostly concern incremental novelty—similar warp-and-inpaint designs have been explored in prior works—and lacking comparisons against modern diffusion-based alternatives. During the rebuttal, the authors addressed the novelty and training concerns, which most reviewers found adequate. While generalization issues remain, consensus formed around the paper’s quality and usefulness.

I hence recommend accepting this submission as a poster.